# CREBAS: Computer-Based REBA Evaluation System for Wood Manufacturers Using MediaPipe

Seong-oh Jeong [1] and Joongjin Kook [2,*]

1    Department of Electronics Information System Engineering, Sangmyung University, 31 Sangmyungdae-gil, Dongnam-gu, Cheonan-si 31066, Chungcheongnam-do, Republic of Korea
2    Department of Information Security Engineering, Sangmyung University, 31 Sangmyungdae-gil, Dongnam-gu, Cheonan-si 31066, Chungcheongnam-do, Republic of Korea
*    Correspondence: kook@smu.ac.kr

**Abstract:** Recently, musculoskeletal disorders (MSDs) caused by repetitive working postures in industrial sites have emerged as one of the biggest problems in the field of industrial health. The risk of MSDs caused by the repetitive working postures of workers is quantitatively evaluated by using NLE (NIOSH Lifting Equation), OWAS (Ovako Working-posture Analysis System), RULA (Rapid Upper Limb Assessment), REBA (Rapid Entire Body Assessment), etc. Methods used for the working posture analysis include vision-based analysis and motion capture analysis. Vision-based analysis is a method where an expert with ergonomics knowledge watches and manually analyzes recorded working images. Although the analysis is inexpensive, it takes a lot of time to analyze. In addition, the analyst's subjective opinions or mistakes may be reflected in the results, so it may be somewhat unreliable. On the other hand, motion capture analysis can obtain more accurate and consistent results, but its measurement equipment is very expensive and it requires a large space for measurement. In this paper, we propose a computer-based automated REBA system that can evaluate, automatically and consistently, working postures in order to supplement the shortcomings of these existing methods. The CREBA system uses the body detection learning model of MediaPipe to detect the worker's area in the recorded images and sets the body area based on the position of the face, detected using the face tracking learning model. In the set area, the positions of joints are tracked using the posture tracking learning model, and the angles of joints are calculated based on the joint positions using the inverse kinematics, and then by automatically calculating the degree of load of the working posture with the REBA evaluation method. In order to verify the accuracy of the evaluation results of the CREBA system, we compared them with the experts' vision-based REBA evaluation results. The result of the experiment showed a slight difference of about 1.0 points between the evaluation results of the expert group and those of the CREBA system. It is expected that the ergonomic analysis method for the working posture used in this study will reduce workers' labor intensity and improve their safety and efficiency.

**Keywords:** REBA; Mediapipe; body pose; wood manufacturer

## 1. Introduction

As the proportion of repetitive tasks increases due to rapid industrial development and the automation of processes, the incidence of musculoskeletal disorders among industrial workers is increasing. In order to reduce occupational musculoskeletal disorders in workers, the Korea Occupational Safety and Health Agency (KOSHA) put industries under the obligation to prevent musculoskeletal disorders from July 2003 (Item 5 of Para. 1 of Art. 24 of the Occupational Safety and Health Act, Chapter 12 of the Rules on Occupational Safety and Health Standards). They issued and have enforced the scope of work burdened by the musculoskeletal system (Ministry of Employment and Labor Notice No. 2003-24), and adopted a method of investigating harmful factors (KOSHA GUIDE H-9-2018), but

patients with musculoskeletal disorders continue to occur [1]. Among domestic industries, musculoskeletal disorders occurring in the top 10 industries with multiple musculoskeletal disorders accounted for 64.9% of all musculoskeletal disorders, and it was found to occur mostly in the manufacturing, wholesale and retail trade, consumer goods repair, and construction [2].

Figure 1 compares the incidence of occupational diseases and the incidence of musculoskeletal diseases based on the annual occurrence of accident cases reported by the Occupational Safety and Health Agency [3]. The report said that, as of 2021, there were 11,868 people suffering from musculoskeletal disorders, accounting for about 87.4% of 13,578 people with occupational diseases, and it has been on a steady rise since 2016. On the other hand, the rate of deaths per 10,000 workers from occupational accidents decreased by 0.22‰ (14%) from 1.25‰ in 2013 to 1.07‰ in 2021, but the musculoskeletal disorders rate per 10,000 workers increased by 2.59‰ (73%) from 3.53‰ in 2013 to 6.12‰ in 2021. According to these statistics, the number of deaths from industrial accidents has decreased, but the number of people with musculoskeletal disorders are on the rise.

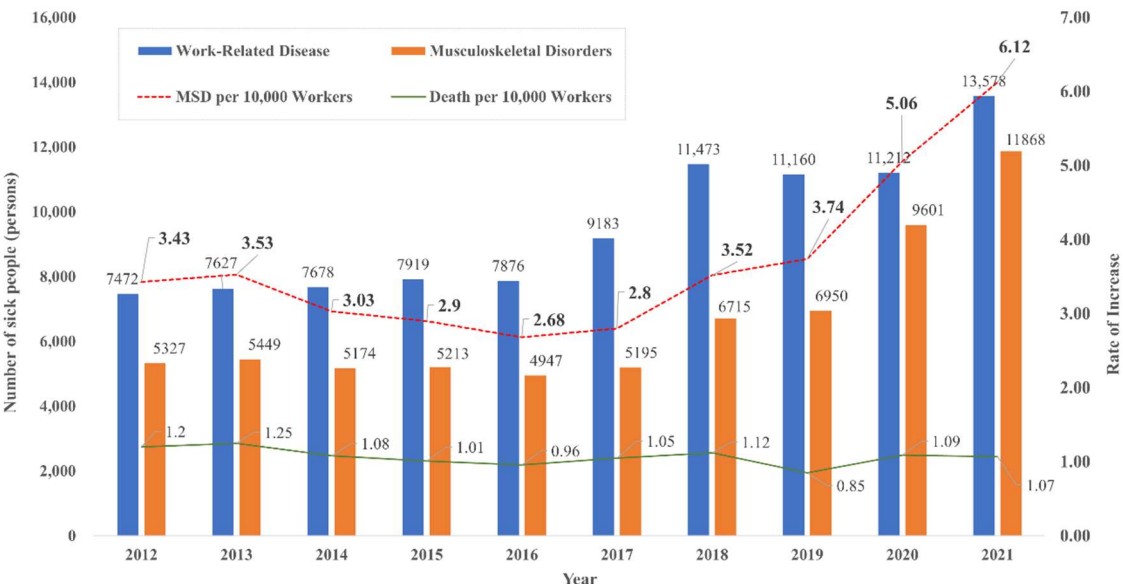

**Figure 1.** Occupational accidents in Korea in 2012–2021.

In particular, as the reported musculoskeletal disorders are mainly caused by repeatedly working in a fixed position, many studies on work improvement and preventive measures have been made in the manufacturing and construction industries. Accordingly, as part of the policy to reduce and prevent musculoskeletal disorders, the Ministry of Employment and Labor legislated the duty of employers to prevent musculoskeletal disorders and made it mandatory to investigate related harmful factors and implement measures to improve the working environment. In addition, it is stipulated that workplace health managers direct the working environments to prevent musculoskeletal disorders by improving work methods and securing ergonomic working spaces. It also specified the role of employers, health managers, and workers, respectively, for preventing musculoskeletal disorders, and the preventive and management measures implemented by characteristics are periodically evaluated and supplemented. However, despite such preventive measures for musculoskeletal disorders at workplace, the frequency is continuously increasing, so the need for discussions on occupational accidents related to musculoskeletal disorders is constantly being raised.

The harmful factor surveys for the prevention of musculoskeletal disorders are conducted at only 11.4%, even in manufacturing workplaces with five or more employees, where people are more likely to have musculoskeletal disorders [4]. In addition, the application of the ergonomic evaluation technique with the objectivity and the quantification

is very low, at about 16%, and the low proficiency of non-experts in using evaluation tools makes the ergonomic evaluation results inaccurate and unreliable. In this study, we proposed a method that uses MediaPipe to track the worker's joint information and automatically calculate the risk through the Rapid Entire Body Assessment (REBA) algorithm. The proposed CREBA method produces more objective and accurate evaluations and lowers the evaluation cost. It aims to design a system that can be easily used by non-experts without ergonomic knowledge by intuitively visualizing data with 3D human body models, along with joint angle and risk factors. This enables small businesses that are difficult to be evaluated by ergonomic experts to conduct the musculoskeletal risk assessment with little cost and effort, and provides ergonomics experts with auxiliary tools for reliable evaluation.

In this study, REBA was selected as an ergonomic precision evaluation tool. REBA is difficult to deduce as a uniform evaluation because many values need to be measured for the whole body, but, unlike other evaluation tools that evaluate only specific areas, it is a suitable evaluation tool for measuring the musculoskeletal workload of manufacturing workers. To compare the accuracy of the evaluation, the evaluation group was composed of three nurses from a Spinal and Joint Hospital with ergonomic expertise. Each evaluator watched a pre-shot work video of a wood worker and compared the results obtained through the ergonomic-based precision evaluation with the evaluation results obtained through this system. Experiments showed an acceptable difference, with a mean bias of score between the evaluation results through the REBA evaluation system proposed in this paper and the evaluation results by ergonomic experts. Therefore, Computer-based Rapid Entire Body Assessment (CREBAS) is expected to be useful for the prevention of musculoskeletal disorders, even in small businesses where it is difficult to obtain a professional evaluation by automatically evaluating the risk of musculoskeletal system of workers.

## 2. Backgrounds

There are two major technologies related to the musculoskeletal risk assessment automation. The first was to evaluate musculoskeletal risk and the evaluation techniques include NIOSH Lifting Equation (NLE), Ovako Working-posture Analysis System (OWAS), Rapid Upper Limb Assessment (RULA), and REBA. Among them, REBA, which deals with the whole body including upper and lower body, arms and wrists, was selected in light of the characteristics of the manufacturing worker. The second is a vision-based posture estimation technique. The depth vision-based posture estimation technique has the highest accuracy among vision-based posture estimation techniques, but the essential use of a depth camera leads to the application of the real field being difficult. Since MediaPipe shows high accuracy for postural estimation, he system was designed using it.

### 2.1. Musculoskeletal Risk Posture Analysis Technique

The KOSHA suggests various precise evaluation methods to evaluate the harmful factors of musculoskeletal disorders of workers, and various studies are also being conducted for this purpose in many other countries. In the study [5], a statistical analysis on 1920 employees of 35 manufacturing companies through cluster sampling was conducted using questionnaires on demographic characteristics, ergonomics-related factors, occupational types, labor intensity, insomnia and Work-related Musculoskeletal Disorders (WMSD), determining the correlation with musculoskeletal disorders. In the study [6], musculoskeletal disorders were evaluated for workers of petrochemical companies. In this study, evaluation was also conducted through a questionnaire. For data analysis, Statistical Package for the Social Sciences (SPSS) was used to evaluate the risk of each job. However, for this evaluation method, the subjective opinion of the evaluator may hamper the ability to draw accurate results, and therefore an expert or systematic objective evaluation method is required.

Precise evaluation of work posture is to determine the risk of musculoskeletal disorders by applying several evaluation techniques based on the posture of the worker, including

OWAS, RULA and REBA. The applied evaluation technique varies depending on the environment or method [7–9].

OWAS is a representative work posture evaluation technique developed by a steel company, Ovako Oy, in Finland in the mid-1970s, and then jointly modified by Ovako Oy and the Finish Institute for Occupational Health. OWAS is widely used because it is easy to learn and easy to apply in the field, but it is difficult to perform detailed analysis with because of the oversimplification of the working posture. The working posture is simply defined into four levels, and the results are also not specific, requiring additional detailed analysis procedures [8,10].

RULA is a working posture assessment technique developed by McAtamney and Corlett at the University of Nottingham, UK, in 1933. It was designed to quickly and easily evaluate the workload caused by working posture, by focusing on the upper limbs such as shoulders, wrists, and neck. RULA helps the use of the EU's minimum safety and health requirements for Visual Display Unit (VDU) workplaces and the UK's guidelines for occupational upper limb disorder prevention. RULA can quickly and easily determine the proportion of workers with upper limbs disorders caused by poor working posture, and it is possible to evaluate the muscle load caused by the task, by examining how working posture affects muscle fatigue, static or repetitive tasks and the force required for the task. Although RULA was developed for providing comprehensive ergonomic evaluation results, the focus is only on the upper limbs, limiting it in evaluating various working postures [8–13].

The REBA assesses an individual worker's exposure to harmful factors associated with musculoskeletal disorders. Compared to RULA, which focuses on upper limbs, the range of the measurement is wide, so it is suitable for analyzing the degree of body burden and exposure to harmful factors in the automobile industry. It supplements the shortcomings of RULA, covering only the upper limbs, by evaluating wrists, forearms, elbows, shoulders, neck, trunk, waist, legs, knees, etc. [8–10,12,14]. Therefore, in order to evaluate the exact workload for the wood manufacturing industry studies in this paper, it is necessary to define the working posture and the appropriate evaluation method.

Table 1 shows the characteristics and the reliability of each evaluation tool. RULA, which has been used in many previous studies, can evaluate the workload on the whole body, but it focuses on the workload on the arm. In the case of OWAS, the whole body is evaluated, but its oversimplification of posture prevent it from subdividing and evaluating various working postures. Lastly, in the case of REBA, although the evaluation reliability is rather low, it has a detailed evaluation system of the whole body, so it is suitable for the evaluation of the whole body workload [13,14]. Therefore, this study intends to evaluate the workload by using REBA, an observation technique suitable for evaluating the workload on the whole body of the worker.

### 2.2. Posture Estimation Techniques

The most commonly used method in vision-based posture estimation technology is a method using depth. Kinect, released by Microsoft in 2010, is the most representative method. Kinect is a device that can extract a person's posture by using a depth information extraction method through an infrared beam projector and a monochrome Complementary Metal-Oxide-Semiconductor (CMOS) sensor. When the infrared laser beam is illuminated, CMOS sensor receives the reflected laser beam points and measures the distance between each pixel, and the image processor processes these data to obtain the user's 3D information. The results of the study [15] verify its usefulness in acquiring the worker's joints with Kinect and Augmented Reality (AR) markers and evaluating the working posture. However, since it is forced to take the depth image with Kinect and to attach AR markers to the worker's body, it is not useful in the real field.

The Forces [16] system enables automated musculoskeletal risk estimation through a workstation featuring motion capture, but it is also difficult to apply to real industrial fields because major joints need to be equipped with markers or sensors for motion capture.

**Table 1.** Characteristics and the reliability of each evaluation tool.

| Evaluation Tool | Characteristics | Evaluation Reliability Standard Deviation t (160) |
|---|---|---|
| OWAS | Easy and simple to apply to the field quickly<br>Difficulty in detailed analysis due to its oversimplification | trunk: 0.96<br>lower arms: 0.95<br>upper arms: 0.17<br>weight: 0.14 |
| RULA | Measure the overall workload, but evaluation is focused on the upper limbs<br>Evaluation accuracy has the highest reliability among the three | ScoreA: 1.13<br>ScoreB: 1.28<br>trunk:1.35<br>neck: 1.35<br>legs: 0.49<br>upper arms: 0.86<br>forearm: 0.73<br>wrist: 0.90<br>wrist twist: 0.26 |
| REBA | Compensate for the shortcomings of RULA confined to the upper limbs<br>Improve the body load measurement | Score A: 1.44<br>Score B: 1.71<br>trunk: 0.73<br>neck: 0.70<br>lower arms: 0.83<br>upper arms: 0.82<br>forearm: 0.50<br>wrist: 0.65 |

Deep neural network learning is called deep learning. Among them, Convolutional Neural Network (CNN) was introduced by LeCun to more effectively process images in deep learning [17], and, later, the recent version CNN was proposed by LeCun in 1998 [18]. Initially, it was used only in the study of the visual cortex of the cerebrum, but since the 1990s, it has been used in image recognition fields. CNN is widely used in image search, autonomous vehicles, and automatic image classification systems, and it is also widely used in other fields, such as speech recognition and natural language processing, as well as the visual fields. Figure 2 shows the CNN architecture designed by LeCun. This architecture consists of convolution layers that extract features, and pooling layers that sample features extracted from the convolution layers. The convolution layers apply a filtering technique to images, and the pooling layers reduce the size of the image by converting local parts of the image into a representative scalar value. The image processing technology using the learning model in this study is based on this CNN.

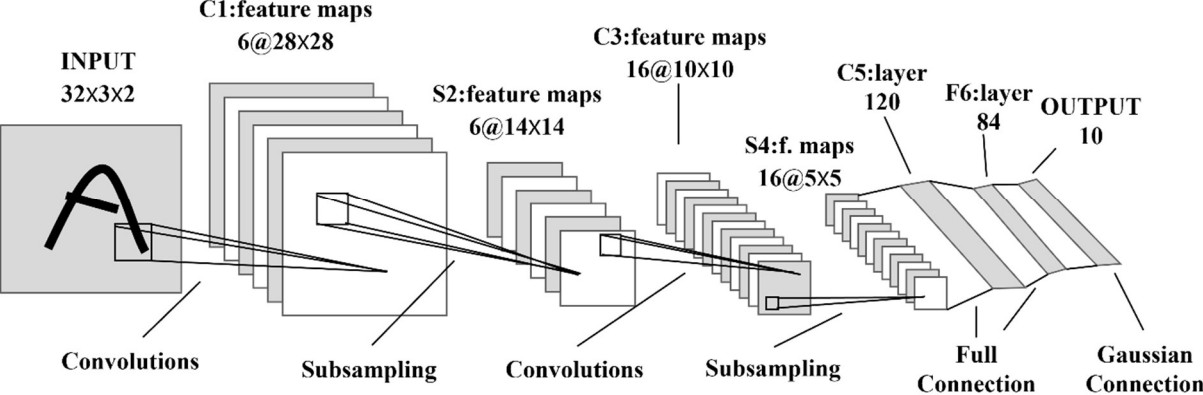

**Figure 2.** Convolutional Neural Network.

OpenPose is the most widely used library for estimating posture by vision. It is a posture estimation model introduced by a team at Carnegie Mellon University (CMU) in 2017, and was released in 2019 with further improvements. OpenPose is a bottom-up model that estimates poses in real time through Part Affinity Fields (PAF) [19].

The posture estimation method is divided into a top-down method and a bottom-up method, depending on which part of the body or which one of the joints is detected first. The top-down method is more accurate than the bottom-up method, but it is slower because it first detects the area of the person in the image, and then estimates the person's posture within their bounding box of. The bottom-up method is less accurate than the top-down method, because it first detects the joints in the image and analyzes the joints' correlations to estimate the posture by connecting them. However, it has the advantage of being fast, because there is no human body tracking process. Figure 3 shows the system architecture of OpenPose [20].

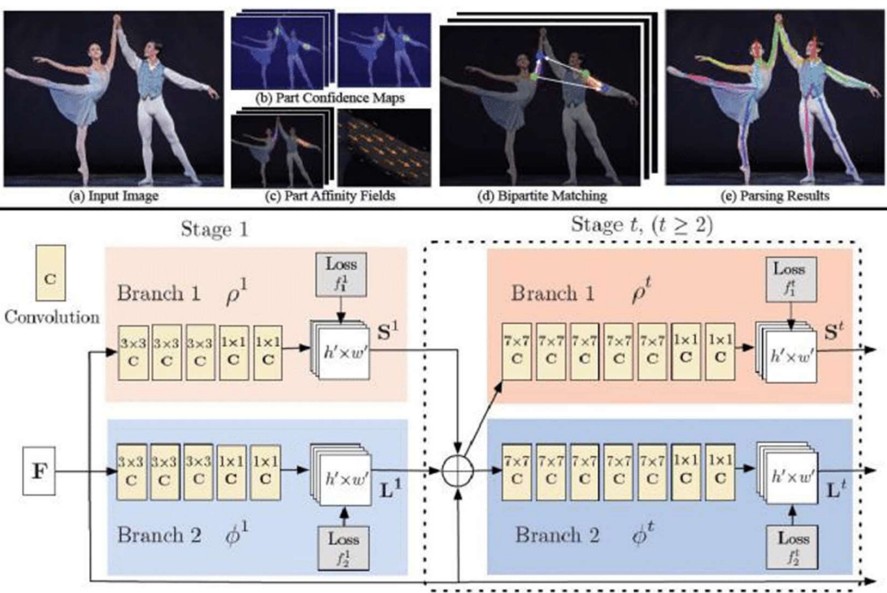

**Figure 3.** OpenPose Architecture. When the image of (**a**) is input, it detects the joint using the VGG-19 model, one of the CNN models, and creates the heatmap of (**b**). Based on the heatmap, the PAF of (**c**) is constructed for the correlation analysis of joints, and the joints are properly divided and connected as shown in (**d**) to obtain the result of (**e**).

OpenPose lacks in accuracy due to this bottom-up method, although its estimation speed is fast. When REBA evaluation is performed using OpenPose, it forces the image to be changed to $432 \times 368$ during the preprocessing, so it could be deformed in the vertical image input, and overlapping or occluded areas could be unmeasured [21].

MediaPipe, another posture estimation technology, is an AI framework provided by Google. It is an open-source cross-platform framework that provides various vision AI functions using various types of perceptual data such as video and audio in the form of a pipeline. This enables a person's face, body parts, and fingers to be estimated in real time. To implement the posture estimation function, which is the key point of this study, requires a model that can know the body skeletal coordinates. The investigation of the characteristics of the above models: they are top-down method, they are light enough to enable real-time inference even with CPU operation, and they can be used for commercial purposes as well. In the posture estimation process, if the user's posture is complicated or occlusion areas occur, the position of the hidden joint needs to be estimated. There are two methods for this: heatmap and regression. The heatmap method is to estimate the position of joints by generating a heatmap for each joint and modifying the offset for each joint. The heatmap method probabilistically calculates the location of major joints among body parts through a learning model, and estimates the position of the most frequent parts in the

form of the heatmap (Figure 4a). Although it has the advantage of being able to recognize the postures of several people, it takes a long time to calculate due to the large amount of computation. On the other hand, the regression model (Figure 4b) requires few computing resources and is highly scalable, but it is vulnerable to occlusion. MidePipe uses a mixture of these two methods to provide a high-accuracy posture estimation with small amount of computation.

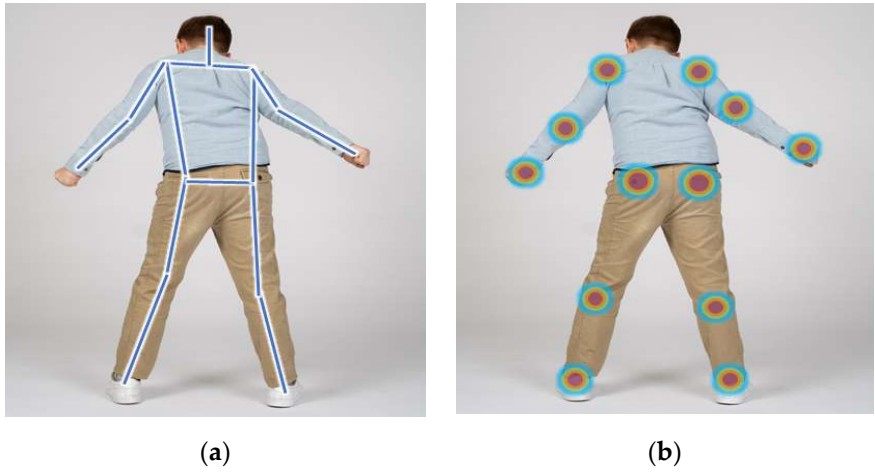

(**a**)                                              (**b**)

**Figure 4.** Posture estimation technique: (**a**) Heatmap; (**b**) Regression.

MediaPipe's pipeline (Figure 5) consists of a body detector and a pose tracker. First, the body area is detected through the body detector, the coordinate of the key-point, and whether there is a person, and the Region of Interest (ROI) of the person is estimated through the posture tracker in the body area. When it is determined that there is a person in the image, only the tracker operates, and when it is determined that there is no person, the detector operates again to detect the body area [22].

Recently, object detection technologies perform the post-processing with Non-Maximum Suppression (NMS). However, NMS can cause malfunctions in complex gestures with overlapping key points, such as hugging or shaking hands. In order to make up for this shortcoming, MediaPipe uses a Face Detector as shown in Figure 6 to detect a face in the image and track the posture based on the face. The face can be quickly estimated because it is the most distinctive among the body parts and has little variation. Based on the position of the face, additional parameters such as the median of the pelvis, the size of the circle including the human body and the inclination depending on the angle of the head are derived.

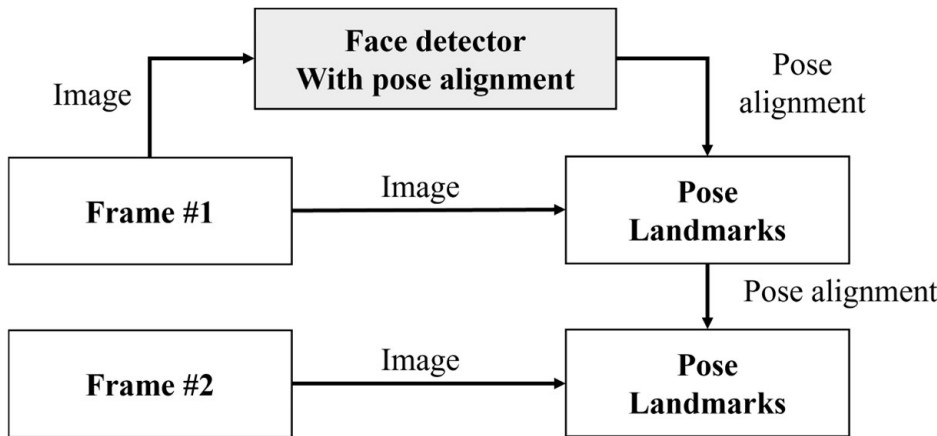

**Figure 5.** MediaPipe pipeline.

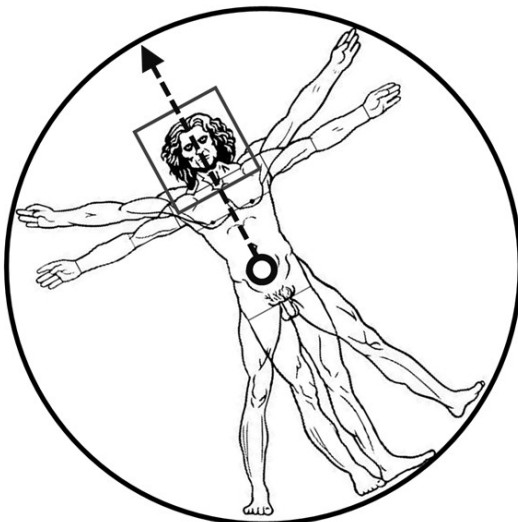

**Figure 6.** Conceptual Diagram of the body area tracking by BlazePose.

The MediaPipe pose pipeline combines key-points used by BlazeFace, BlazePalm, and Coco to track 33 joint key-points (See Figure 7). Unlike OpenPose or *Kinect*, it estimates only the minimum key-points necessary to estimate the rotation and size ROI positions [23].

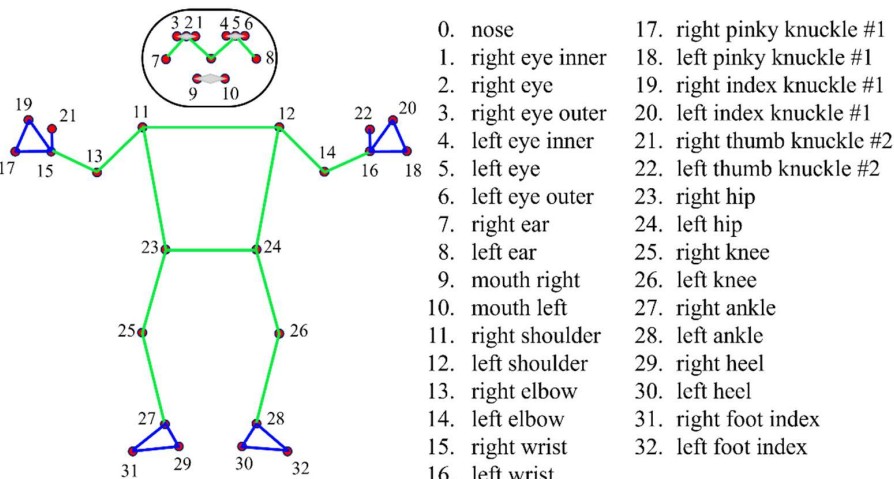

**Figure 7.** MediaPipe Pose Landmark.

In this study, a posture estimation technique that can accurately estimate the posture of a worker in real time is required. The research analyzing the performance of OpenPose and MediaPipe compared two datasets consisting of 1000 images, each of which includes one or two people [24]. The first AR dataset consisted of a variety of people and backgrounds without a specific topic, while the second, yoga, dataset consisted only of yoga and fitness postures. For consistent comparison, only 17 joints commonly used in OpenPose and MediaPipe were used for evaluation. The dataset used as the evaluation index has a Percent of Correct Points (PCK) of 20% tolerance, and the BlazePose model, a posture model used in MediaPipe, was compared by capacity (Full, Lite). BlazePose Full is a high-capacity model used in a desktop, and BlazePose Lite is a low-capacity model used in a mobile smartphone. As shown in the comparison results in Table 2, OpenPose had the highest accuracy in the AR dataset and BlazePose Full showed the highest accuracy in the yoga dataset. As for the frame rate, BlazePose Full was 25 times faster than OpenPose in a desktop environment (20 Core CPU), and BlazePose Lite was 70 times faster than OpenPose in a mobile environment (Google Pixel).

**Table 2.** Frame rate comparison between OpenPose and MediaPipe with AR dataset.

| Model | FPS | AR Dataset, PCK@0.2 | Yoga Dataset, PCK@0.2 |
|---|---|---|---|
| OpenPose (CPU) | 0.4 | 87.8 | 83.4 |
| BlazePose Full | 10 | 84.1 | 84.5 |
| BlazePose Lite | 31 | 79.6 | 77.6 |

Table 3 shows the Frame Per Second (FPS) results when compared using the experimental data of this study: the image of a wood manufacturing worker. The hardware used in the experiment is CPU: AMD Ryzen 7 2700X, GPU: NVIDIA GeForce RTX 2070. As a result of the experiment, MediaPipe was at an average speed of 30 fps with CPU operation, and OpenPose showed slow performance with an average 0.3 fps for CPU operation and 10 fps for GPU operation despite high-performance PC environment.

**Table 3.** The FPS comparison of the posture of a wood manufacturing worker.

| MediaPipe (CPU) | OpenPose (CPU) | OpenPose (GPU) |
|---|---|---|
| 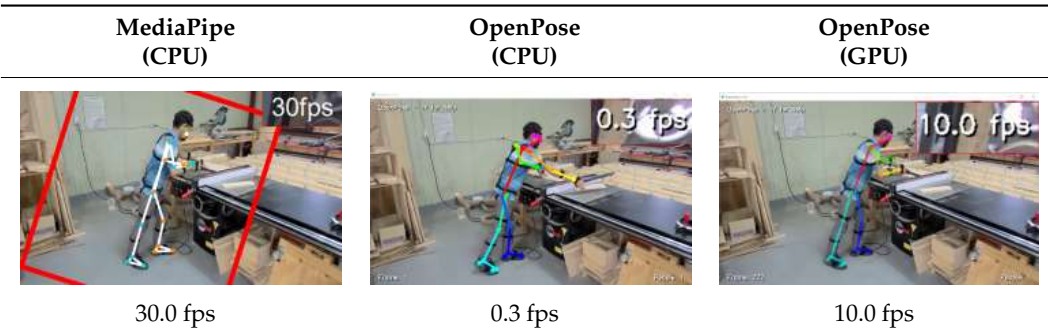 | | |
| 30.0 fps | 0.3 fps | 10.0 fps |

In general, considering whole-body posture images that do not take into account the occlusion and the intersection of body parts, OpenPose is more accurate than MediaPipe, but it is significantly slower in speed. Because it is necessary to estimate the posture of a worker in real time in this study, MediaPipe was used.

As for the video (iPhone11 Pro shot, 1080p resolution) of the working posture of a wood manufacturing worker used as experimental data in this study, the worker's posture was estimated normally in the range of 2.5 to 6.0 m between the worker and the camera (See Figure 8).

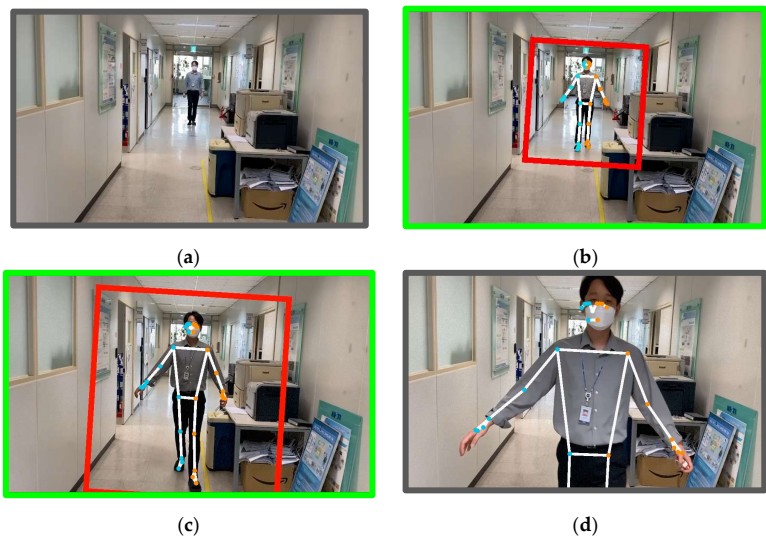

**Figure 8.** Posture detection and estimation based on the distance: (**a**) 7.0 m; (**b**) 6.0 m; (**c**) 2.5 m; (**d**) 1.0 m.

## 3. REBA Evaluation System Using MediaPipe

The evaluation system proposed in this paper uses Unity3D, a 3D rendering engine, as an authoring tool to handle the worker's joints in a 3D virtual environment [25]. Figure 9 shows the architecture of this evaluation system. First, it detects a body area from an image input through MediaPipe and extracts a 3D joint landmark by tracking the posture in the detected area. The measured joint information is visualized in three dimensions through a virtual skeletal model. Based on the relative position of each joint, the joint angle is calculated and input into the REBA evaluation module.

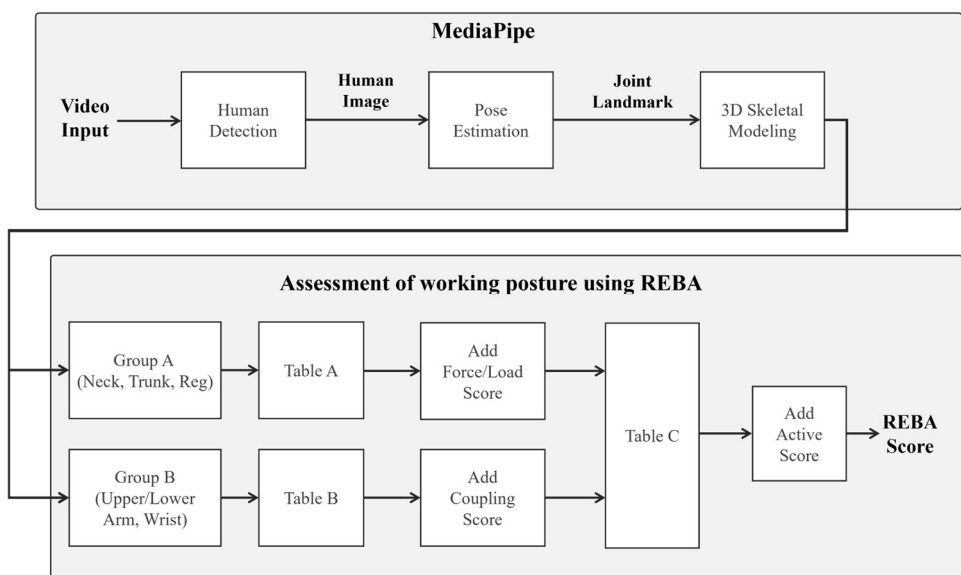

**Figure 9.** REBA evaluation system using MediaPipe.

### 3.1. System Design

Figure 10 shows the Graphic User Interface (GUI) of this evaluation system. For visualization, the basic GUI provided by Unity3D was used. Expressing the detected body areas and joints in the shape of a skeleton on the input image enhanced its visibility. Figure 10 shows the overall GUI of the system. For GUI visualization, image, text, toggle, and button, which are GUI components provided by Unity3D, were used, and Render Texture and Line Render were used to visualize the detected areas and 3D skeleton models.

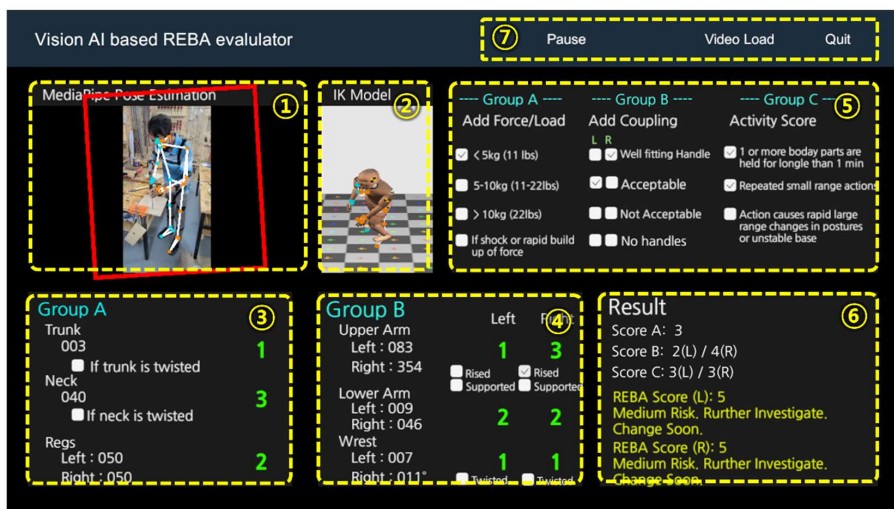

**Figure 10.** GUI of REBA evaluation system using MediaPipe.

- Visualization of Human Pose Estimation results: The working posture image is encoded in RGB32 format and 640 × 480 resolution and output through the Image UI. The Bounding Box of the body area obtained by MediaPipe's Human Detector and the joint obtained by Pose Tracking is overlaid on the image.
- Visualization of Inverse Kinematics (IK) modeling: The position coordinates of each joint calculated through Human Pose Estimation are input to the target of the inverse kinematics (IK) model, and the applied result is displayed.
- Details of Group A (waist, neck, legs): The bending angles of the hip, neck, and leg joints included in group A are displayed in white, and the scores for each part are displayed in green based on the REBA rule.
- Details of Group B (upper arm, forearm, wrist): The bending angles of the shoulder, elbow, and wrist joints included in group B are displayed in white, and the scores for each part are displayed in green based on the REBA rule.
- Selection of additional risk factors: Select additional deduction factors based on the group A's weight/force classification, group B's grip type, and group C's behavioral score.
- Final results (score A, score B, score C, REBA score): The result of group A is score A, the result of group B is score B, and the sum of score A and score B is score C. The final REBA score is displayed in yellow by each left body and right body.
- Top menu (pause, image selection, end): The pause button is to stop the video being played, and the image selection button is to call up another image. The end button is to end the program.

### 3.2. Joint Position Estimation

The Tensorflow-based BalzeFace learning model detected the face area from the input video image. The body area was visualized by finding the body center-point based on the face position and by drawing a rectangular boundary based on the face position and center-point (Figure 11).

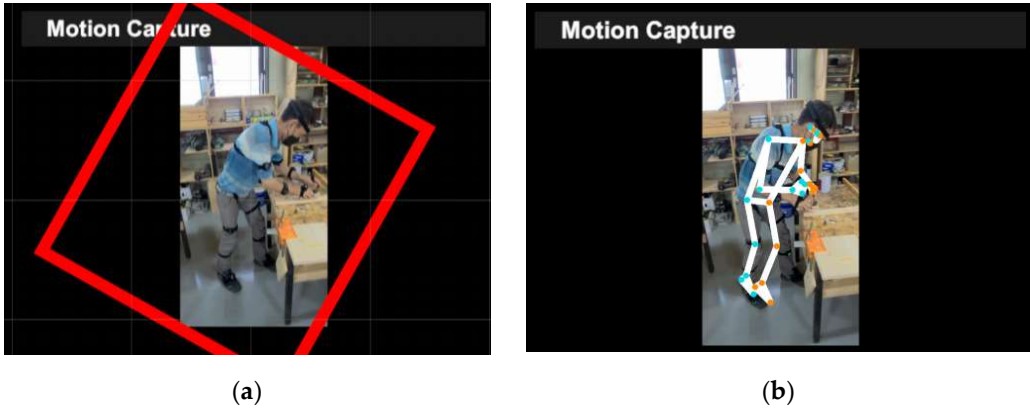

(**a**)　　　　　　　　　　　　　　　　　　　　　　　　（**b**)

**Figure 11.** Detection and body estimation: (**a**) Visualization of the detected body area; (**b**) body estimation and visualization.

As shown in Figure 7, 32 joint landmark values are calculated through the BlazePose [26] learning model by inputting the Crop Texture of the body region areas through human body detection. The position and connection information of each joint were visualized with points and lines.

### 3.3. Calculation of the Angles of Joints Using Inverse Kinematics (IK)

The input data required for the REBA evaluation method are the angles of six joints: waist, neck, leg, upper arm, forearm, and wrist. However, the joint landmark calculated by MediaPipe does not include the waist and neck, and it has only coordinated information, not angle.

Equation 1 is to find the midpoint M between two connected points to obtain the position of the waist and neck. The position of the pelvis is the center of the left hip (Landmark 23) and the right hip (Landmark 24). The position of the chest is the center of the left shoulder (Landmark 11) and the right shoulder (Landmark 12), and these two positions enable us to know the position of the waist. Equation (1) is to find the midpoint M of two points in a three-dimensional space, and Figure 12 shows the concept for this equation.

$$M = \left( \frac{x^1 + x^2}{2}, \frac{y^1 + y^2}{2}, \frac{z^1 + z^2}{2} \right) \tag{1}$$

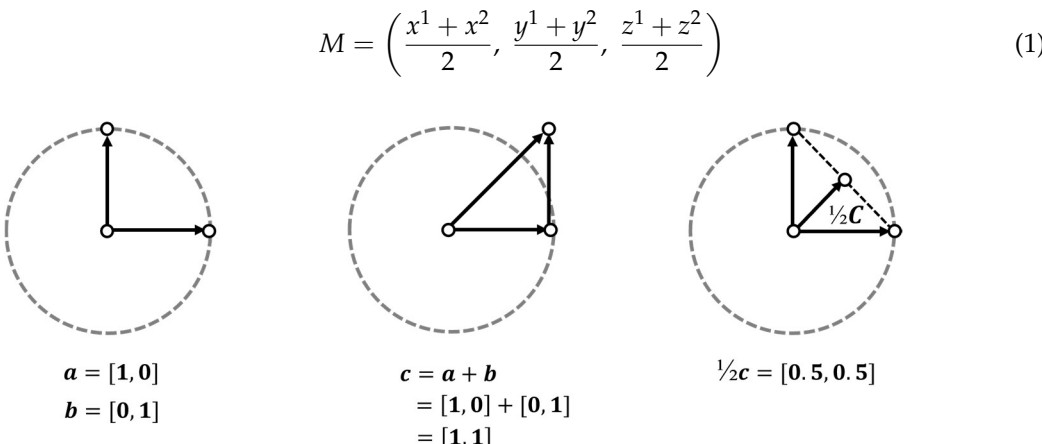

**Figure 12.** Procedure of finding midpoints of two points.

Inverse Kinematics (IK) was used to obtain the joint angles required for REBA evaluation. IK is mainly used in the computer animation and robotics fields and is the opposite of Forward Kinematics. In Forward Kinematics, when the position and direction of the higher-level object in hierarchy change, the position and direction of the lower-level object are affected and determined; on the contrary, in Inverse Kinematics, the position and direction of child object affects the position and direction of parent object.

In this study, Final IK provided by Unity Asset Store was used for IK calculation [27]. Final IK provides Full Body IK based on Cyclic Coordinate Descent (CCD) [28]. CCD is suitable for calculating the Inverse Kinematics of joints with complex structures. Figure 13 shows the process of CCD. Assume there are five joints, and when $P_1$ is the Base Joint and E is the End Effector, the angles of all joints are sequentially changed so that the difference in distance and direction between the End Effector and the Target (T) is minimized.

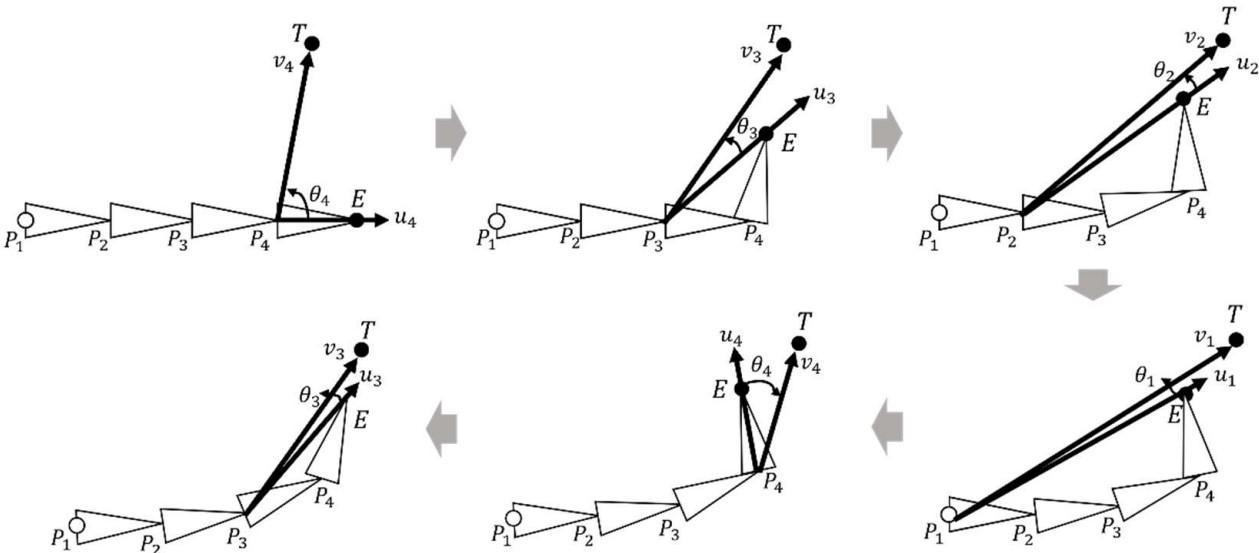

**Figure 13.** Procedure of CCD IK.

Equation (2) is to find the angle of each joint through the dot product of two vectors (u, v) obtain from the Target position and the positions of the two joints.

$$\theta_i = \cos^{-1}\left(\frac{u_i \cdot v_i}{|u_i||v_i|}\right) \tag{2}$$

Figure 14 shows a model with Full Body IK applied, and the ball on the right hand of the model is an object for the calculation for the right-hand joint. Based on the IK, the position of the ball object appropriately changes the positions of the rest of the joints, creating a natural posture.

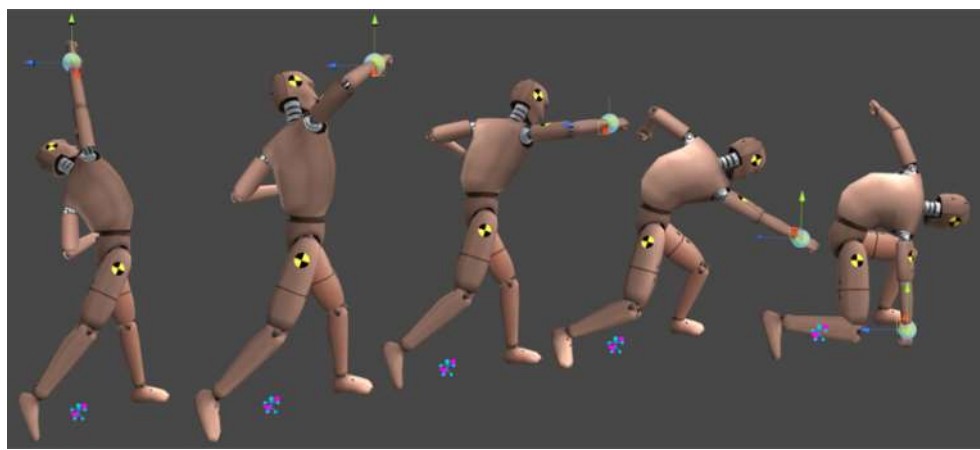

**Figure 14.** The position change of joints depending on the position change of the target.

Figure 15 is a human body model in which body parts are connected in a hierarchical form. The Human IK model is basically configured in a top-down manner based on the spine. The model is divided into shoulder and hip based on spine. The shoulder consists of head, arm, and hand, and the hip consists of leg, knee, and foot.

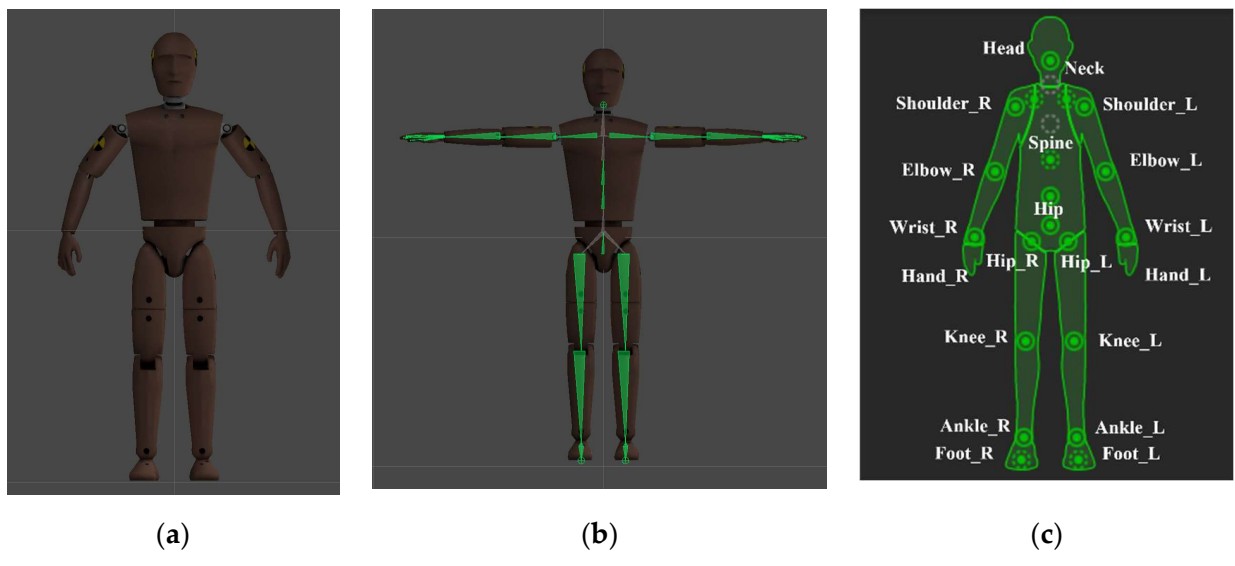

(a)　　　　　　　　　　　　(b)　　　　　　　　　　　　(c)

**Figure 15.** Avatar model information to which IK will be applied: (**a**) Avatar rigging model; (**b**) Avatar skeleton mapping; (**c**) Joint point information.

As shown in Figure 16, the joint landmark of MideaPipe and the Scale Factor of the IK model were matched, and a total of nine joints were linked to an End Effector. Inputting the maximum weight of all linked effectors increases the response sensitivity of the connected joints. Figure 16 shows the angle GUI of the joint through the linked IK model.

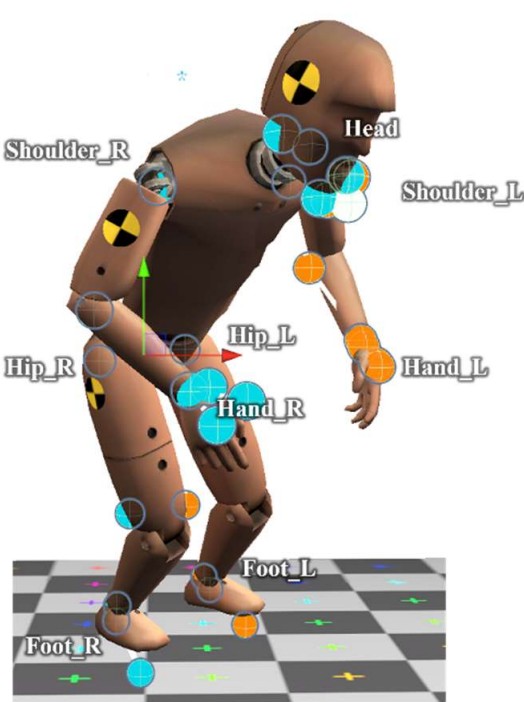

**Figure 16.** Angle GUI of the joint through the linked IK model.

*3.4. REBA Posture Evaluation Algorithm*

The working posture evaluation procedure of REBA is shown in Figure 17. In order to evaluate the working posture, the load value for the working posture needs to be calculated. Scores are given depending on the angles of the skeletons belonging to each group, and the load values are calculated using the given scores [8,9,12,14].

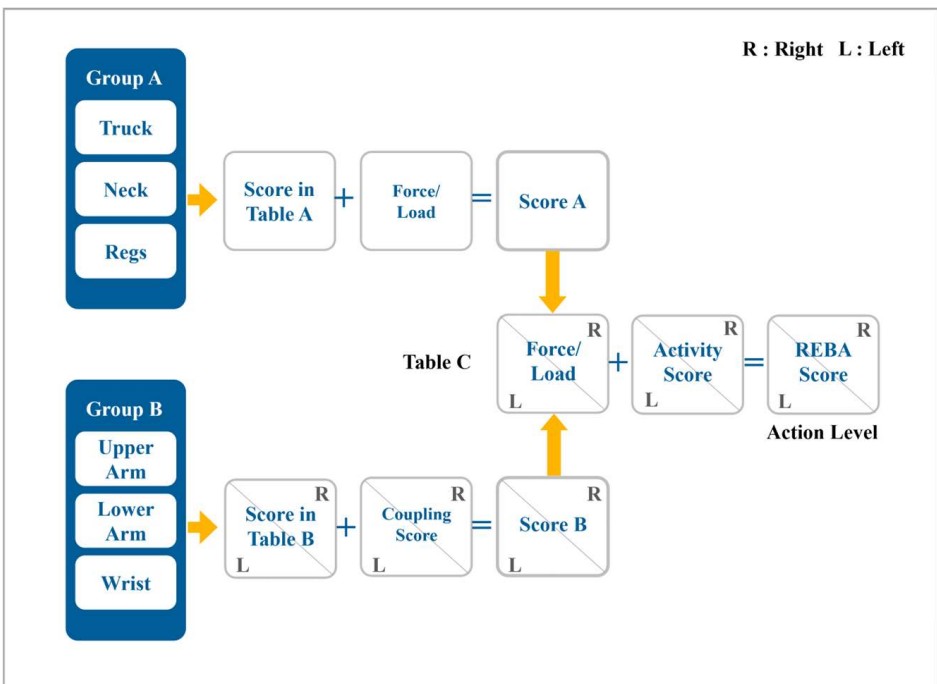

**Figure 17.** REBA evaluation procedure.

3.4.1. Group A Evaluation

First, the parts belonging to Group A are trunk, waist, neck, and legs. Scores for each posture are calculated based on the posture of each part, and then adjusted depending on

body twisting or lateral bending. Scores are calculated based on the classification system of body, neck, and leg postures in Tables 4–6. One point is added with twisting or side bending in the case of neck and waist, and one point or two points are added depending on the knee bending angle in the case of legs.

**Table 4.** Classification based on the body posture.

| Index | Working Posture |
|-------|-----------------|
| 1 | Upright posture |
| 2 | 0~20° bending or 0~20° reclining |
| 3 | 20~60° bending or more than 20° reclining |
| 4 | More than 60 bending° |
| +1 | Trunk is twisted or bent sideways |

**Table 5.** Classification based on the neck posture.

| Index | Working Posture |
|-------|-----------------|
| 1 | 0~20° bending |
| 2 | More than 20° bending |
| +1 | Neck is twisted or bent sideways |

**Table 6.** Classification based on the leg posture.

| Index | Working Posture |
|-------|-----------------|
| 1 | Both legs kept side by side or walking/sitting |
| 2 | Only one foot is supported on the ground |
| +1 | Knee bent 30° to 60° |
| +2 | Knee bent more than 60° |

Since the scores determined by the classification system represent the individual load level of each part, it is necessary to combine them. Therefore, using REBA Table A in Table 7, the scores of the three parts are combined, and the evaluation of the handling weight in Table 8 is added.

**Table 7.** Scoreboard for REBA Table A (waist, neck, legs).

| Neck | Legs | Waist | | | | |
|------|------|-------|---|---|---|---|
| | | 1 | 2 | 3 | 4 | 5 |
| 1 | 1 | 1 | 2 | 2 | 3 | 4 |
| | 2 | 2 | 3 | 4 | 5 | 6 |
| | 3 | 3 | 4 | 5 | 6 | 7 |
| | 4 | 4 | 5 | 6 | 7 | 8 |
| 2 | 1 | 1 | 3 | 4 | 5 | 6 |
| | 2 | 2 | 4 | 5 | 6 | 7 |
| | 3 | 3 | 5 | 6 | 7 | 8 |
| | 4 | 4 | 6 | 7 | 8 | 9 |
| 3 | 1 | 3 | 4 | 5 | 6 | 7 |
| | 2 | 3 | 5 | 6 | 7 | 8 |
| | 3 | 5 | 6 | 7 | 8 | 9 |
| | 4 | 6 | 7 | 8 | 9 | 9 |

**Table 8.** Evaluation of handling weight.

| <5 kg | 5–10 kg | >10 kg | Shock or Sudden Force |
|-------|---------|--------|-----------------------|
| 0 | 1 | 2 | +1 |

3.4.2. Group B Evaluation

As for Group B, scores are calculated based on the posture of the upper arm, lower arm, and wrist, and the score is adjusted depending on the posture of the upper arm and wrist as shown in Tables 9–11. In the case of the upper arm, one point is added when arm is open or rotated, or the shoulder is lifted, and one point is decreased when the arm is supported by something. In tfhe case of the wrist, a twist adds one point.

**Table 9.** Classification based on the upper arm posture.

| Index | Working Posture |
|---|---|
| 1 | 20° reclining or 20° lifting forward |
| 2 | Reclining more than 20° or 20~45° lifting forward |
| 3 | 45~90° lifting forward |
| 4 | More than 90° lifting forward |
| +1 | Upper arm is stretched or rotated |
| +1 | Shoulders lifted |
| -1 | Arm is supported or leaned on something |

**Table 10.** Classification based on the lower arm posture.

| Index | Working Posture |
|---|---|
| 1 | 60~100° lifting |
| 2 | More than 100° lifting or 0~60° lifting |

**Table 11.** Classification based on the wrist.

| Index | Working Posture |
|---|---|
| 1 | 0~15° bending or lifting |
| 2 | More than 15° bending or lifting |
| +1 | Wrist is twisted |

The score determined by the classification system of upper arm, lower arm, and wrist posture is combined using REBA Table B in Table 12, and the evaluation of the handle in Table 13 is added.

**Table 12.** Scoreboard for REBA Table B (upper arm(shoulder), lower arm, wrist).

| Lower Arm | Wrist | Upper Arm (Shoulder) | | | | |
|---|---|---|---|---|---|---|
| | | 1 | 2 | 3 | 4 | 5 |
| 1 | 1 | 1 | 2 | 2 | 3 | 4 |
| | 2 | 2 | 3 | 4 | 5 | 6 |
| | 3 | 3 | 4 | 5 | 6 | 7 |
| 2 | 1 | 1 | 3 | 4 | 5 | 6 |
| | 2 | 2 | 4 | 5 | 6 | 7 |
| | 3 | 3 | 5 | 6 | 7 | 8 |

**Table 13.** Classification of handle types.

| Good | Acceptable | Bad | Very Bad |
|---|---|---|---|
| With a strong and well-fixed handle located in the center of gravity | With acceptable handle or if a part of the object can be used like a handle | Not suitable to hold by hand even though it can be lifted or with an inappropriate handle | No handle or with a dangerous type of handle |

### 3.4.3. Determination of the Overall Workload

Finally, the posture score A and the posture score B are combined using REBA Table C in Table 14, and, additionally, the behavior scores in Table 15 are added to calculate the REBA score. The calculated REBA score is taken another action through the decision-making right in Table 16.

**Table 14.** Scoreboard for REBA Table C.

| | | Posture Score A | | | | | | | | | | | |
|---|---|---|---|---|---|---|---|---|---|---|---|---|---|
| | | **1** | **2** | **3** | **4** | **5** | **6** | **7** | **8** | **9** | **10** | **11** | **12** |
| Posture score B | 1 | 1 | 1 | 2 | 3 | 4 | 6 | 7 | 8 | 9 | 10 | 11 | 12 |
| | 2 | 1 | 2 | 3 | 4 | 4 | 6 | 7 | 8 | 9 | 10 | 11 | 12 |
| | 3 | 1 | 2 | 3 | 4 | 4 | 6 | 7 | 8 | 9 | 10 | 11 | 12 |
| | 4 | 2 | 3 | 3 | 4 | 5 | 7 | 8 | 9 | 10 | 11 | 11 | 12 |
| | 5 | 3 | 4 | 4 | 5 | 6 | 8 | 9 | 10 | 10 | 11 | 12 | 12 |
| | 6 | 3 | 4 | 5 | 6 | 7 | 8 | 9 | 10 | 10 | 11 | 12 | 12 |
| | 7 | 4 | 5 | 6 | 7 | 8 | 9 | 9 | 10 | 11 | 11 | 12 | 12 |
| | 8 | 5 | 6 | 7 | 8 | 8 | 9 | 10 | 10 | 11 | 12 | 12 | 12 |
| | 9 | 6 | 6 | 7 | 8 | 9 | 10 | 10 | 10 | 11 | 12 | 12 | 12 |
| | 10 | 7 | 7 | 8 | 9 | 9 | 10 | 11 | 11 | 12 | 12 | 12 | 12 |
| | 11 | 7 | 7 | 8 | 9 | 9 | 10 | 11 | 11 | 12 | 12 | 12 | 12 |
| | 12 | 7 | 8 | 8 | 9 | 9 | 10 | 11 | 11 | 12 | 12 | 12 | 12 |

**Table 15.** Static/repetitive behavior scores.

| Index | Behavior Scores |
|---|---|
| +1 | If one or more body parts are held (ex: hold for more than a minute) |
| +1 | Repetitive tasks in a narrow range (ex: repeat more than 4 times per minute except for walking) |
| +1 | Rapidly changing behavior over a wide range or unstable lower body posture |

**Table 16.** REBA decision-making right.

| Level | REBA Score | Risk Level | Measurement (Further Investigate) |
|---|---|---|---|
| 0 | 1 | Very low | No change |
| 1 | 2–3 | Low | Change may be needed |
| 2 | 4–7 | Medium | Change soon |
| 3 | 8–10 | High | Investigate and change soon |
| 4 | 11–15 | Very high | Implement change |

## 4. Experiments and Results

In order to verify the accuracy of the system, the seven types of postures (cutting machines, circular saws, drilling machines, saws, chisels, planers, loading/unloading of heavy objects) in Figure 18 that are most commonly used in actual wood and wood product manufacturing plants were selected, and the working scenes were recorded as videos. Based on the recorded images, the accuracy was evaluated by comparing the evaluation results of experts with the results of the proposed system.

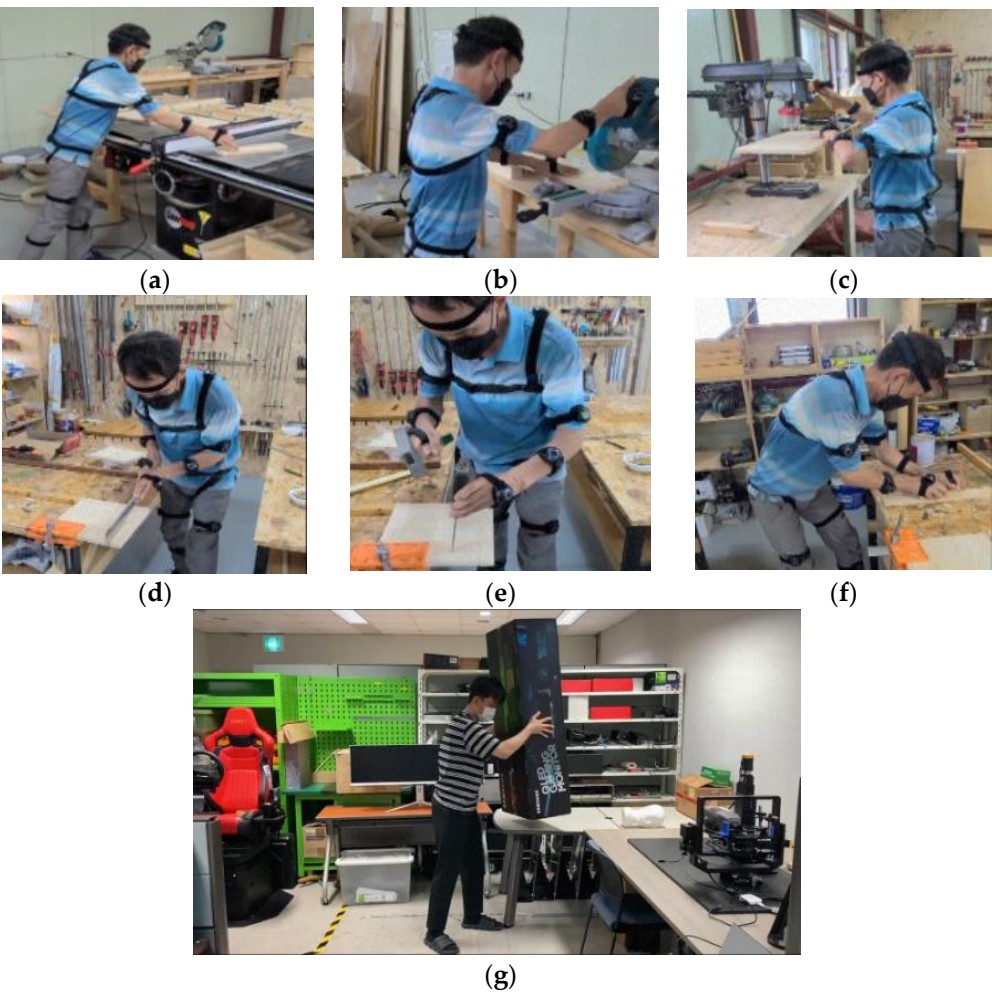

**Figure 18.** Working posture depending on wood processing tools: (**a**) cutting machines; (**b**) circular saws; (**c**) drilling machines; (**d**) saws; (**e**) chisels; (**f**) planers; (**g**) loading/unloading of heavy objects.

### 4.1. Experimental Environments

The precision evaluation experiment in this study consists of three stages (evaluation preparation, expert evaluation, and feedback). First, the consent of ergonomic experts to participate in the experiment was obtained, and detailed information about the task to be evaluated (work description, photos, weight of the task object, number of repetitions of the task) was provided. Second, ergonomics experts watched the recorded video and evaluated it by applying REBA. Finally, feedback was provided on the results, in which errors occurred compared to the results of this system.

The six working posture images were shot with the iPhone 11 Pro's default camera, and the resolution of the images was 1080p. During shooting, the distance between the worker and the camera was set to 2.5~3.0m, and the face was made to appear on the screen (Figure 19).

### 4.2. Evaluation Results by REBA Experts

The Golden Reference for the task to be evaluated in this study was determined by three ergonomic experts (nurses at a spine and joint hospital). The participating ergonomic experts independently performed the ergonomic precision evaluation (REBA), shown in Figure 20, through six images, and the average value of the calculated evaluation results was determined as the golden reference.

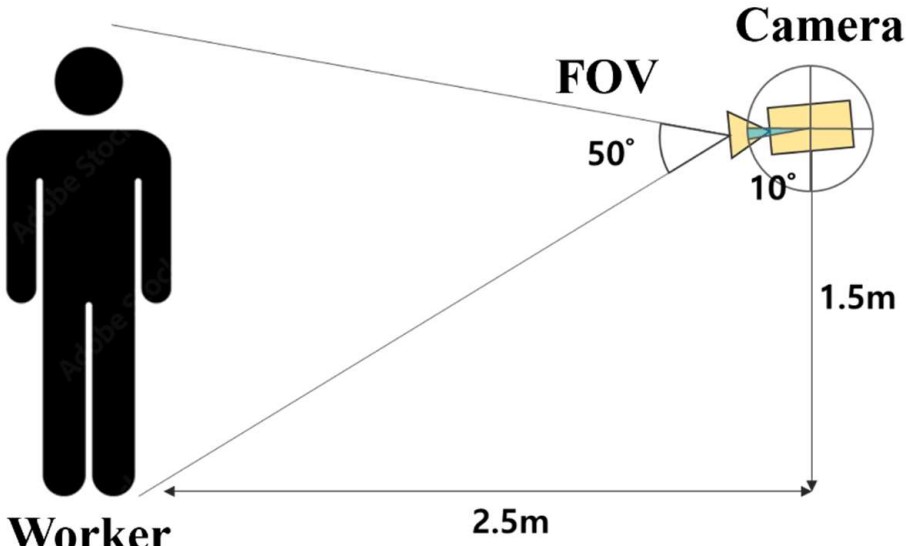

**Figure 19.** Camera angle and distance.

### 4.2.1. Evaluation of Cutting Machine Working Posture

Figure 21 shows the results of the REBA evaluation of three evaluators on the working posture using the cutter. The left and right parts of group B (upper arm, forearm, wrist) were calculated as average values, and two behavioral points were added because of the fixed body parts and repeated work in a narrow range, resulting in 7 points, 6 points, and 5.5 points. As a result, the level of the action was found to be Level 2 (Further Investigate). In detail of the cutting machine working posture, the score A was high with 2.5 points, 4 points, and 5 points, respectively, and it was found that the waist and leg posture scores were a big part.

### 4.2.2. Evaluation of Circular Saw Working Posture

Figure 22 shows the results of the REBA evaluation of three evaluators for the circular saw working posture. The left and right parts of group B (upper arm, forearm, and wrist) were calculated as average values, and two behavioral points were added because of the fixed body parts and repeated work in a narrow range, resulting in three points, four points, and five points. As a result, the level of action was found to be Level 1 (Change may be needed) and Level 2 (Further investigate/Change Soon). In detail of the circular saw working posture, the score B was slightly higher than that of other skeletons, with two points, two points and three points, but the overall score was good and it is considered to be a working posture with low musculoskeletal risk.

### 4.2.3. Evaluation of Drilling Machine Working Posture

Figure 23 shows the results of the REBA evaluation of three evaluators on the drilling machine working posture. The left and right parts of group B (upper arm, forearm, wrist) were calculated as average values, and two behavioral points were added because of the fixed body parts and repeated work in a narrow range, resulting in three points, four points, and five points. As a result, the level of the action was found to be Level 1 (Change may be needed) and Level 2 (Further Investigate/Change Soon). In detail of drilling machine working posture, the score B was found to be high, with 1.5 points, 4 points, and 5.5 points, and it was found that the shoulder score in particular was a big part.

| working details | | | working posture | | |
|---|---|---|---|---|---|
| bend the body and hold the plywood with the legs apart for balance sa as to push and cut it | | |  | | |
| trunk (waist) | | 2 | upper arms (shoulder) | L | R |
| | | | | 2 | 1 |
| standing posture | 1 | ·twist: +1 ·bending sideways: +1 | >20˚ lifting or tilting | 1 | ·pronate/supinate: +1 ·lifting: +1 ·supported: −1 |
| 0˚–20˚ bending 0˚–20˚ stretching | 2 | | 20˚–45˚ lifting >−20˚ tilting backward | 2 | |
| 20˚–60˚ bending >20˚ stretching | 3 | | 45˚–90˚ lifting forward | 3 | |
| >60˚ bending | 4 | | >90˚ lifting forward | 4 | |
| neck | | 1 | lower arms | L | R |
| | | | | 2 | 2 |
| 0˚–20˚ bending | 1 | ·twist: +1 ·bending sideways: +1 | 60˚–100˚ bending | 1 | |
| >20˚ bending or stretching | 2 | | <60˚ or >100˚ beding | 2 | |
| legs | | 2 | wrist | L | R |
| | | | | 2 | 1 |
| walking or sitting posture with weight on both sides | 1 | ·knee bending 30˚-60˚: +1 ·knee bending over 60˚ : +2 | 0˚–15˚ bending/ stretching | 1 | ·rotation/twist: +1 |
| walking with weight on one side or unstable posture | 2 | | >15˚bending/ stretching | 2 | |
| score A (Table A) | | 3 | score B (Table B) | L | R |
| | | | | 3 | 1 |
| weight/force | | 0 | handle | L | R |
| | | | | 1 | 1 |
| <5Kg | 0 | shock/rapid build up of force: +1 | good | 0 | |
| 5–10Kg | 1 | | acceptable | 1 | |
| >10Kg | 2 | | bad | 2 | |
| | | | very bad | 3 | |

| Score A + Score B = Score C (Table C) | L | 4 | R | 3 |
|---|---|---|---|---|

| behavioral scores | 1 or more body parts are held , ex) holding for more than 1min. | +1 |
|---|---|---|
| | repeated small range actions, ex) repeat more than 4 times per min (escept for walking) | +1 |
| | action causes rapid large range changes in postures or unstable base | +1 |

| REBA SCORE (L) | 6 | Risk Level (L) | Medium (Further Investigate, Change Soon) |
|---|---|---|---|
| REBA SCORE (R) | 5 | Risk Level (R) | Medium (Further Investigate, Change Soon) |

**Figure 20.** Examples of filling out the evaluation sheet.

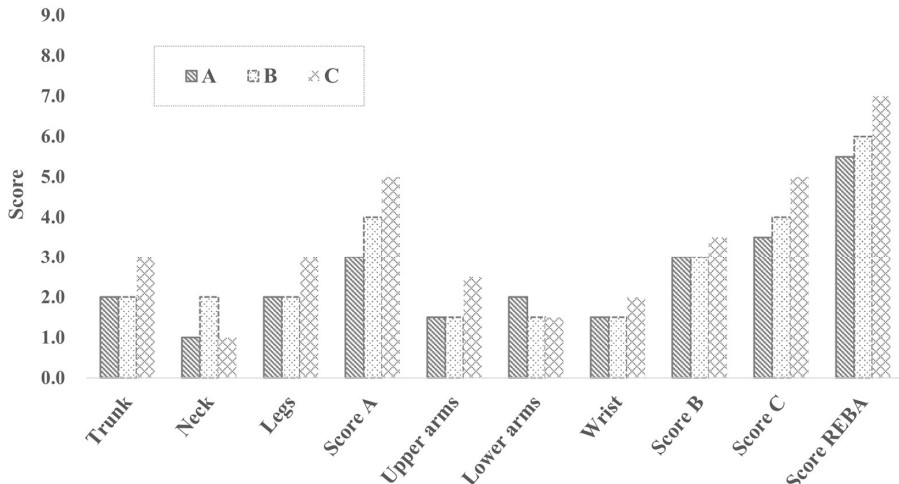

**Figure 21.** Precision evaluation results of cutter working posture.

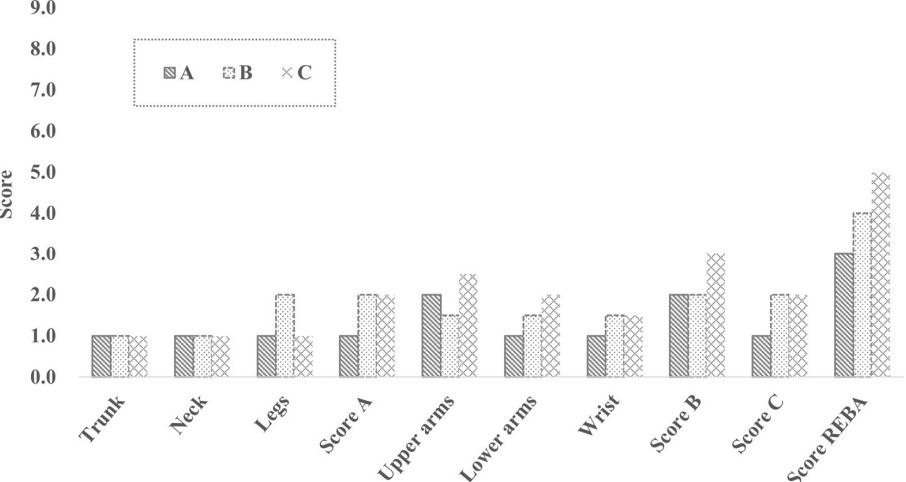

**Figure 22.** Precision evaluation results of circular saw working posture.

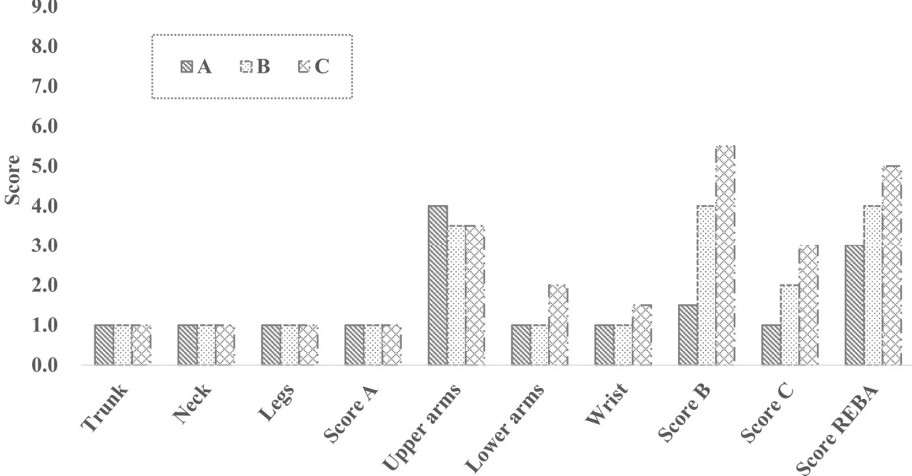

**Figure 23.** Precision evaluation results of drilling machine working posture.

4.2.4. Evaluation of Saw Working Posture

Figure 24 shows the results of the REBA evaluation of three evaluators for the saw working posture. The left and right parts of group B (upper arm, forearm, and wrist) were

calculated as average values, and two behavioral points were added because of the fixed body parts and repeated work in a narrow range, resulting in 5.5 points, 6 points, and 4 points. As a result, the level of action was found to be Level 2 (Further investigate/Change Soon). In detail of saw working posture, the score A was high, with four points, three points, and five points, and it was found that the scores of waist, neck and legs were, in particular, a big part.

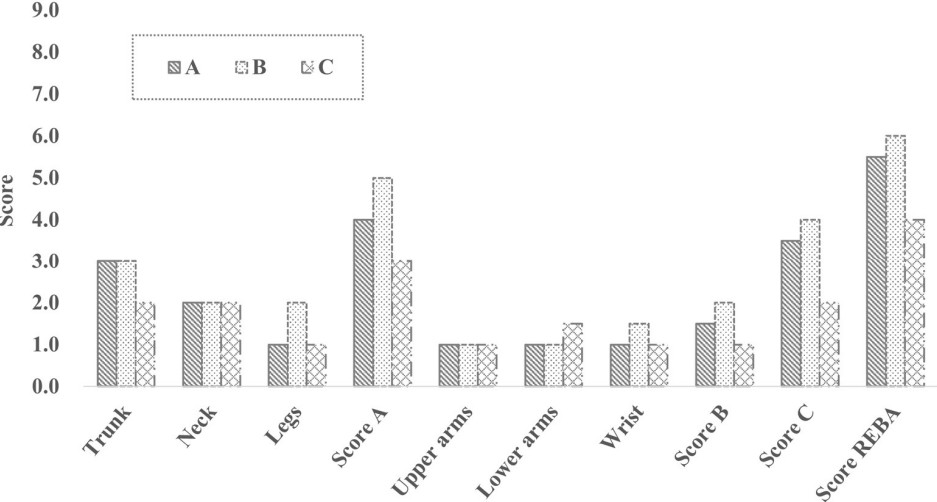

**Figure 24.** Precision evaluation results of saw working posture.

### 4.2.5. Evaluation of Chisel Working Posture

Figure 25 shows the results of the REBA evaluation of three evaluators for the chisel working posture. The left and right parts of group B (upper arm, forearm, and wrist) were calculated as average values, and two behavioral points were added because of the fixed body parts and repeated work in a narrow range, resulting in six points, six points, and five points. As a result, the level of action was found to be Level 1 (Change may be needed). In detail of chisel working posture, the score A was high, with four points, four points, and two points, and it was found that the scores of waist, neck and were, in particular, a big part.

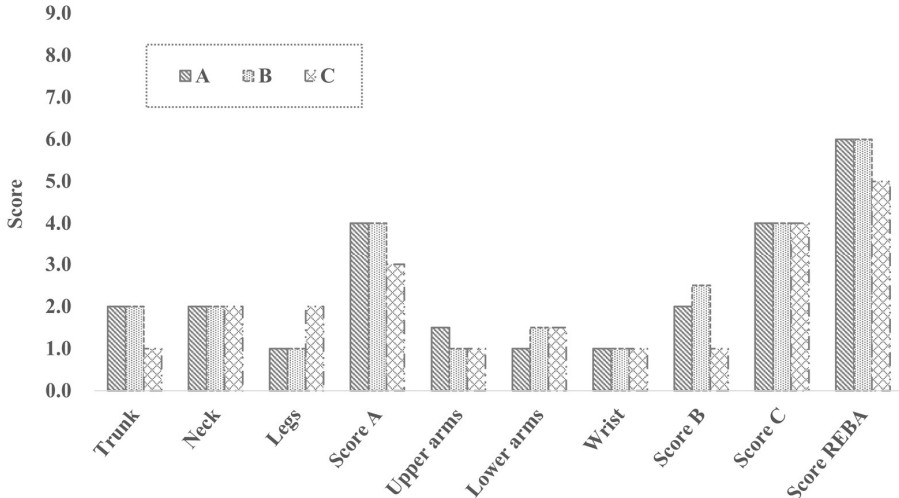

**Figure 25.** Precision evaluation results of chisel working posture.

### 4.2.6. Evaluation of Planer Working Posture

Figure 26 shows the results of the REBA evaluation of three evaluators for the planer working posture. The left and right parts of group B (upper arm, forearm, and wrist)

were calculated as average values, and two behavioral points were added because of the fixed body parts and repeated work in a narrow range, resulting in 6 points, 8 points, and 4.5 points. As a result, the level of action was found to be Level 2 Medium Risk (Further investigate/Change Soon) and Level 3 High Risk (Investigate/Implement Change). In detail of planer working posture, the scores of waist, neck and shoulder were high.

### 4.2.7. Evaluation of the Posture for Heavy Load Work

Figure 27 shows the results of the REBA evaluation of three evaluators for the heavy load working posture. The left and right parts of group B (upper arm, forearm, and wrist) were calculated as average values, and two behavioral points were added because of the fixed body parts and repeated work in a narrow range, resulting in 13 points, 10.5 points, and 10 points. As a result, the level of action was found to be Level 4 Very High Risk (Implement Change). In detail of heavy load working posture, the scores of waist, neck and shoulder were very high.

### 4.3. Evaluation Results by Computer-Based REBA

The seven working postures (cutters, circular saws, drilling machines, saws, chisels, planers, and heavy loads) used in the REBA evaluation using MediaPipe are commonly repeated postures in the wood manufacturing industry. In the evaluation system, the REBA score of the worker's motion and posture is indicated.

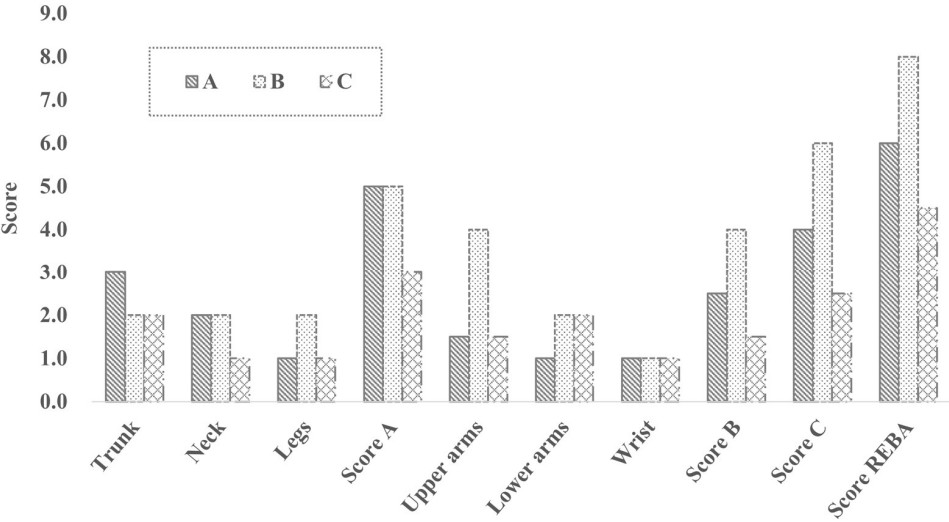

**Figure 26.** Precision evaluation results of planer working posture.

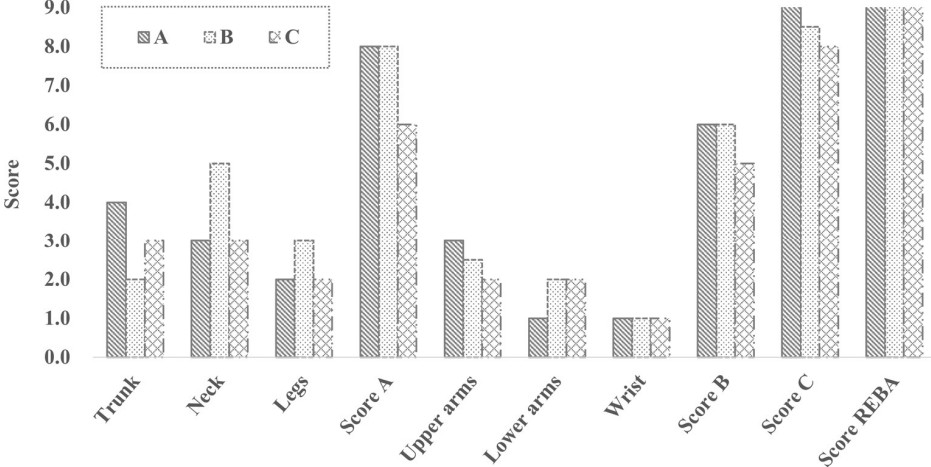

**Figure 27.** Precision evaluation results of heavy load working posture.

### 4.3.1. Evaluation of Cutting Machine Working Posture

Posture analysis was conducted by inputting the same working posture image used for expert evaluation into the vision AI-based REBA evaluation system developed through this study. Figure 28 is the result of evaluating the cutting machine working image with the Vision AI-based working posture evaluation system. The bending score of the waist and legs was good with one point, but the bending score of the neck was high with three points. On the other hand, the results of the shoulder, elbow, and wrist were generally good. Combining the results of groups A and B, the score C was low with three points and two points on the left and right, but because more than one body part is fixed and repetitive tasks are performed in a narrow range, two behavioral points were added. The final REBA score was five on the left and four on the right (Medium risk, Further Investigate).

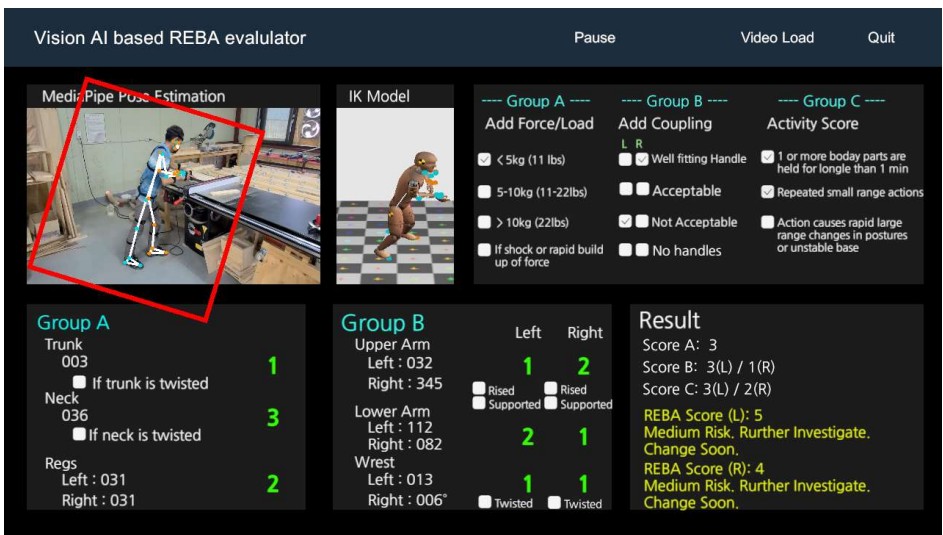

**Figure 28.** Evaluation of cutting machine working posture.

### 4.3.2. Evaluation of Circular Saw Working Posture

Figure 29 is the result of evaluating the circular saw machine working image. The joint scores were good overall, but the risk score of the right shoulder was high due to the lifting posture. Because more than one body part is fixed and repetitive tasks are performed in a narrow range, two behavioral points were added. The final REBA evaluation results are five points on both sides (Medium Risk, Further Investigate/Change soon).

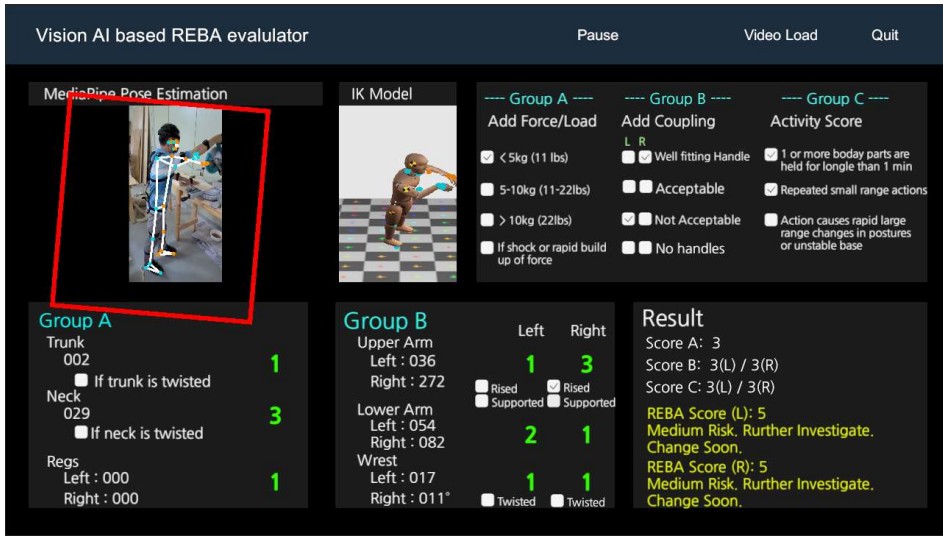

**Figure 29.** Evaluation of circular saw working posture.

### 4.3.3. Evaluation of Drilling Machine Working Posture

Figure 30 is the result of evaluating the drilling machine working image. The joint scores were good overall, but the elbow angle was not good and the risk score of the right shoulder was high due to the lifting posture. Because more than one body part is fixed and repetitive tasks are performed in a narrow range, two behavioral points were added. The final REBA evaluation result is three points on both sides (Low Risk, Change may be needed).

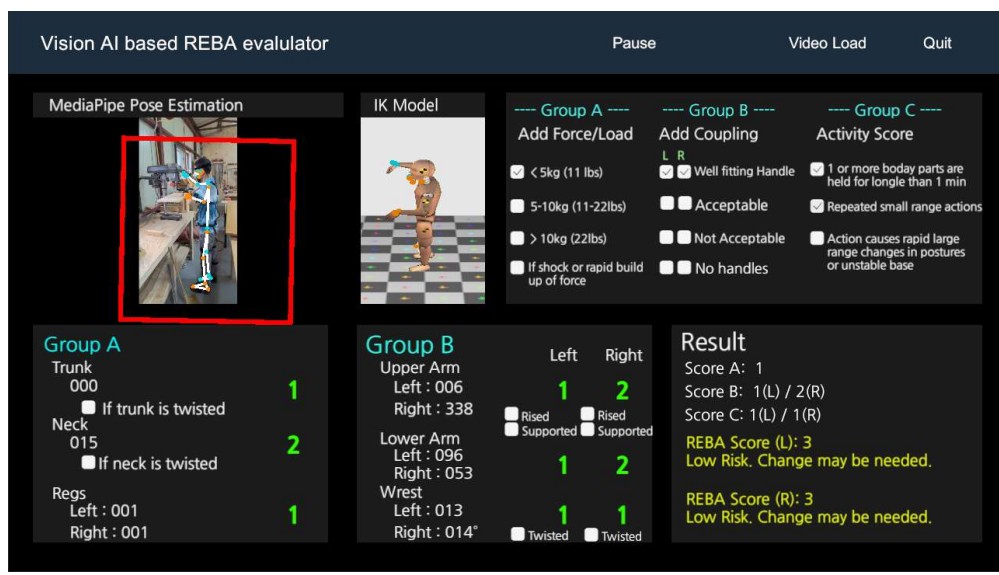

**Figure 30.** Evaluation of drilling machine working posture.

### 4.3.4. Evaluation of Saw Working Posture

Figure 31 is the result of evaluating the saw working image. The lower back, neck, and legs were higher due to the lower back posture. The risk scores of shoulder, elbow, and wrist were good, but because more than one body part is fixed and repetitive tasks are performed in a narrow range, two behavioral points were added. The REBA evaluation results are five points on both sides (Medium Risk, Further Investigate/Change Soon).

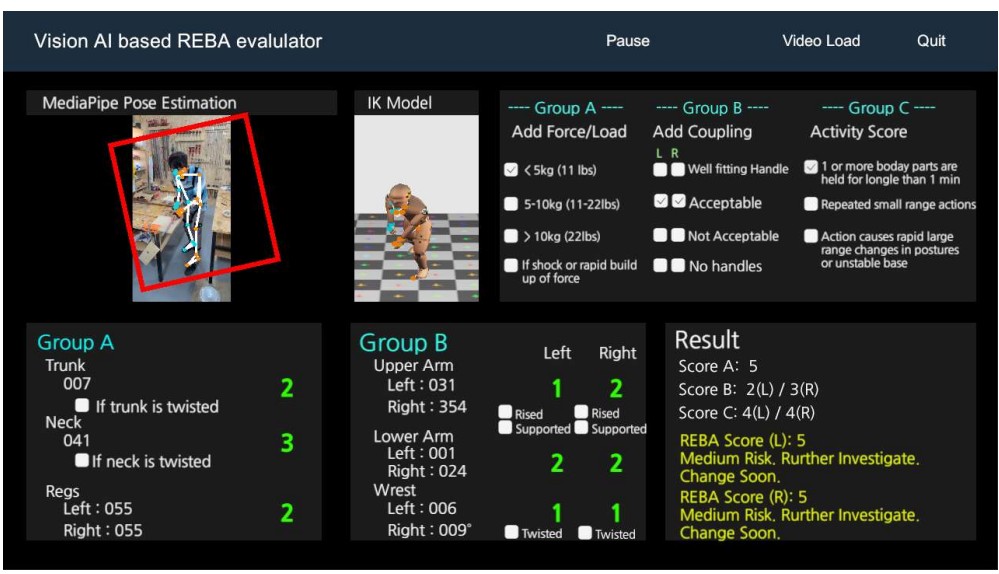

**Figure 31.** Evaluation of saw working posture.

### 4.3.5. Evaluation of Chisel Working Posture

Figure 32 is the result of evaluating the chisel working image. The risk scores of waist, neck, and leg were slightly higher due to the posture of bending the waist and head, and the strength/weight score was added by one point due to hammering. In addition, since more than one body part is fixed and repetitive tasks are performed in a narrow range, two behavioral points were added. The final REBA results were six points on the left and seven points on the right (Medium Risk, Further Investigate/Change Soon).

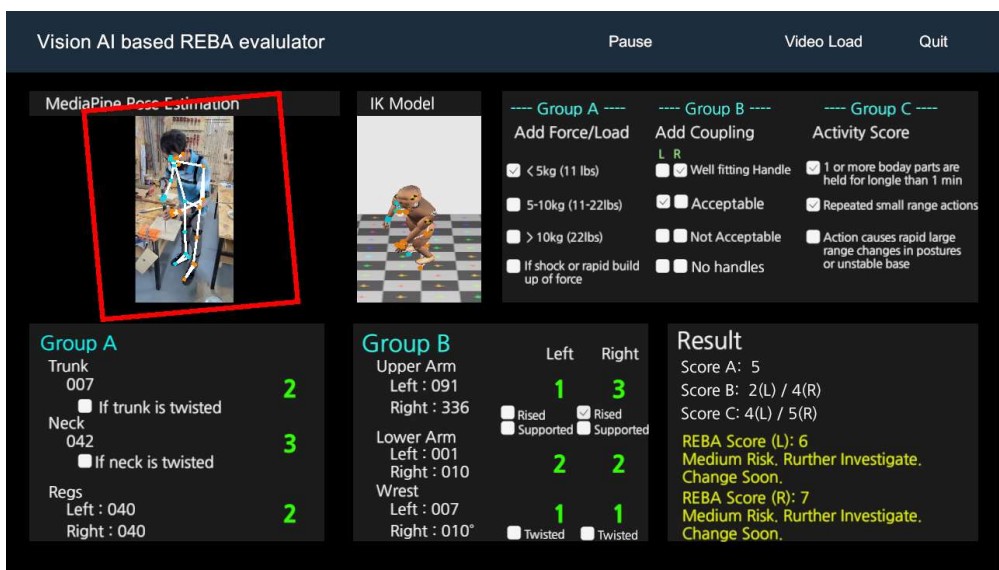

**Figure 32.** Evaluation of chisel working posture.

### 4.3.6. Evaluation of Planer Working Posture

Figure 33 is the result of evaluating the planer working image. The risk scores of wrist, neck, and leg were rather high due to the posture of bending the body and the head. In addition, more than one body part is fixed and repetitive tasks are performed in a narrow range, so two behavioral points were added. The final REBA results were seven points on the left (Medium Risk, Further Investigate/Change Soon) and eight points on the right (High Risk, Investigate and Implement Change).

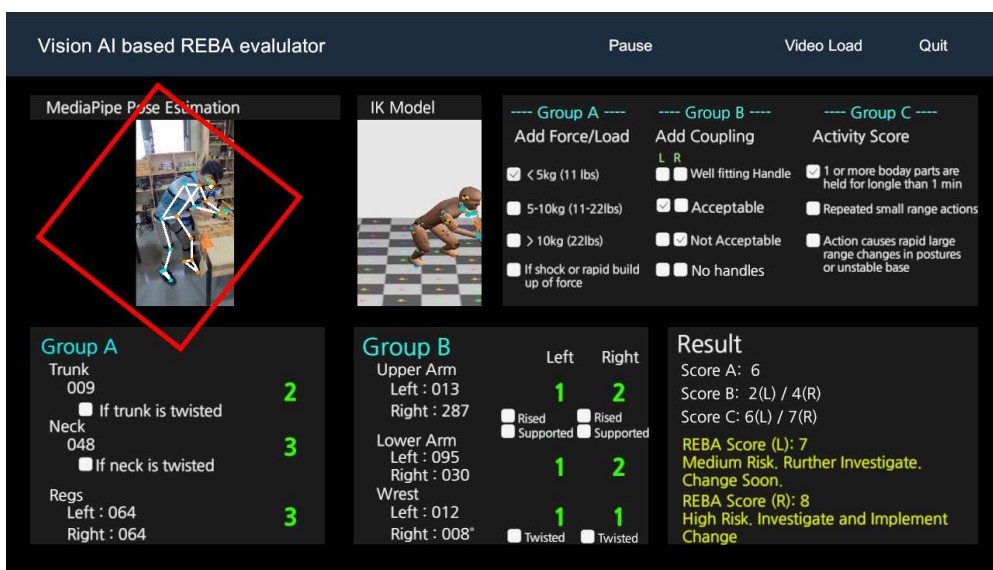

**Figure 33.** Evaluation of planer working posture.

### 4.3.7. Evaluation of the Posture for Heavy Load Work

Figure 34 is the result of evaluating the image of the heavy load working posture. The scores were high overall due to the posture of bending the body sideways; the head and the legs also greatly influenced the results. The weight of the load was 20 kg, and two points were added to the weight score, and the handle score was high because there was no grip. In addition, since more than one body part is fixed, and repetitive tasks are performed in a narrow range, two behavioral points were added. The final REBA results were 10 points on the left (High Risk, Investigate and Implement Change) and 11 points on the right (High Risk, Investigate and Implement Change).

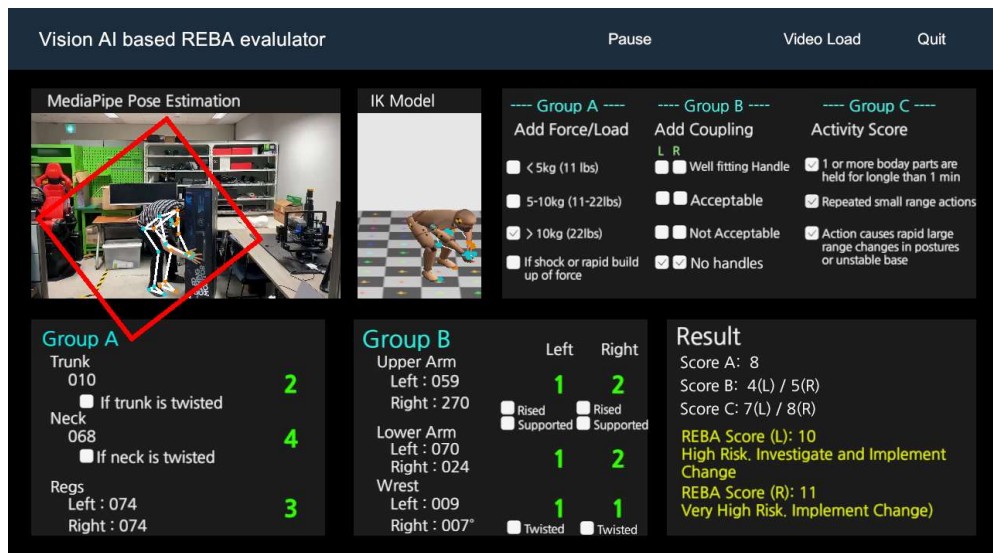

**Figure 34.** Evaluation of the posture for heavy load work.

### 4.4. Comparison of Evaluation Results

REBA evaluation by experts shows differences depending on the proficiency of the evaluator. In addition, it takes about 20 min to explain the REBA evaluation method and the evaluation video to the subject, and about 10 to 15 min to fill out the evaluation table. On the other hand, in the CREBA evaluation, the average time was measured by performing the evaluation 100 times for each of the seven postures, and the results are shown in Table 17.

**Table 17.** Evaluation time required for each posture of CREBA.

| Type | Data Size (MB) [1] | Minimum | Maximum | Average |
|---|---|---|---|---|
| Cutter | 6.51 | 1 | 1 | 2.6 |
| Circular saw | 4.98 | 1 | 1 | 2 |
| Drilling machine | 5.11 | 1.3 | 1 | 1.3 |
| Saw | 6.76 | 0.3 | 0 | 0 |
| Chisel | 6.16 | 1.6 | 1 | 4 |
| Planer | 4.93 | 1 | 2.6 | 1 |
| Heavy load | 8.68 | 1.3 | 1.3 | 1.3 |

[1] All videos are encoded at $1280 \times 720$, 30 fps.

To verify the accuracy of the method proposed in this study, the system implemented in the selected working posture and the evaluation results of the evaluator were compared. Table 18 shows the average scores of three evaluators for six types of work postures (cutting machine, circular saw, drilling machine, saw, chisel, and planer), and Table 19 is the evaluation scores using MediaPipe.

**Table 18.** REBA evaluation results by experts (average).

| Group | Body Part | Cutter | Circular Saw | Drilling Machine | Saw | Chisel | Planer | Heavy Load |
|---|---|---|---|---|---|---|---|---|
| A | Waist | 2.3 | 1 | 1 | 2.6 | 1.6 | 2.3 | 3 |
| | Neck | 1.3 | 1 | 1 | 2 | 2 | 1.6 | 3.6 |
| | Legs | 2.3 | 1.3 | 1 | 1.3 | 1 | 1.3 | 2.3 |
| | Weight | 0 | 0.3 | 0 | 0 | 1 | 1 | 2 |
| | Score A | 4 | 1.6 | 1 | 4 | 3.3 | 3.6 | 7.3 |
| B | Upper Arms | 2 | 1 | 2.6 | 1 | 0.6 | 2.6 | 3 |
| | Lower Arms | 1.6 | 1.3 | 1.3 | 1.3 | 1 | 1.6 | 1 |
| | Wrist | 2 | 1 | 1 | 1.3 | 1 | 1 | 1 |
| | Handle | 1.3 | 1 | 1 | 1.3 | 0.6 | 1.3 | 3 |
| | Score B (Left) | 4 | 2 | 3.6 | 0.6 | 1.6 | 4 | 3 |
| | Upper Arms | 1.6 | 3 | 3.3 | 1 | 1.6 | 2.3 | 3 |
| | Lower Arms | 1.6 | 1.6 | 1.3 | 1 | 1.6 | 1.6 | 1 |
| | Wrist | 2 | 1.6 | 1.3 | 1 | 1 | 1 | 1 |
| | Handle | 0.6 | 0 | 0 | 0 | 0.6 | 1 | 3 |
| | Score B (Right) | 2.3 | 4 | 3.6 | 1 | 2 | 3.3 | 3 |
| REBA | Left | 6.6 | 3.6 | 3.6 | 5.3 | 5 | 7 | 13 |
| Score | Right | 5.6 | 4.3 | 4 | 5 | 5 | 6.6 | 13 |

**Table 19.** REBA evaluation results by CREBA (average).

| Group | Body Part | Cutter | Circular Aw | Drilling Machine | Saw | Chisel | Planer | Heavy Load |
|---|---|---|---|---|---|---|---|---|
| A | Waist | 1 | 1 | 1 | 2 | 2 | 2 | 2 |
| | Neck | 3 | 2 | 2 | 3 | 3 | 3 | 5 |
| | Legs | 2 | 1 | 1 | 2 | 2 | 2 | 3 |
| | Weight | 0 | 1 | 0 | 0 | 1 | 0 | 2 |
| | Score A | 3 | 2 | 1 | 5 | 6 | 5 | 8 |
| B | Upper Arms | 1 | 1 | 1 | 1 | 1 | 1 | 2 |
| | Lower Arms | 1 | 1 | 2 | 2 | 2 | 1 | 2 |
| | Wrist | 1 | 1 | 1 | 1 | 1 | 1 | 1 |
| | Handle | 1 | 1 | 1 | 0 | 0 | 1 | 2 |
| | Score B (Left) | 2 | 2 | 2 | 1 | 1 | 2 | 5 |
| | Upper Arms | 2 | 3 | 3 | 2 | 2 | 2 | 3 |
| | Lower Arms | 1 | 1 | 2 | 2 | 2 | 1 | 2 |
| | Wrist | 1 | 1 | 1 | 1 | 1 | 1 | 1 |
| | Handle | 0 | 0 | 0 | 0 | 0 | 1 | 2 |
| | Score B (Right) | 1 | 3 | 4 | 2 | 2 | 2 | 7 |
| REBA | Left | 5 | 3 | 3 | 6 | 6 | 6 | 10 |
| Score | Right | 4 | 3 | 4 | 6 | 6 | 6 | 11 |

Overall, the comparison of the cutting machine working posture scores (Figure 35) showed similar results between the evaluation scores of the ergonomic expert and the evaluation scores of the proposed system. As for the scores for each joint, the MediaPipe scores were lower in all parts except the neck, and the part showing the biggest difference was the trunk, with an error of 1.3 points. The REBA scores were 4.5 points and 6.7 points, respectively, indicating the same action level (Level 2).

The comparison of the circular saw working posture scores (Figure 36) showed similar results overall. The scores were the same for the trunk and upper arms, and the MediaPipe scores were lower in other parts except the neck. The part showing the biggest difference was the neck, with an error of one point. The REBA scores were three points and four points, respectively, indicating the action level of Level 2 and Level 3.

As for the comparison of drilling machine working posture scores (Figure 37), the scores for each joint were the same for the trunk, legs, and wrist, and in other parts except for the neck and lower arms, where the MediaPipe scores were lower. The part showing the biggest difference was the upper arms, with an error of 1.5 points. The REBA scores were 3.5 points and 4 points, indicating the same action level (Level 2).

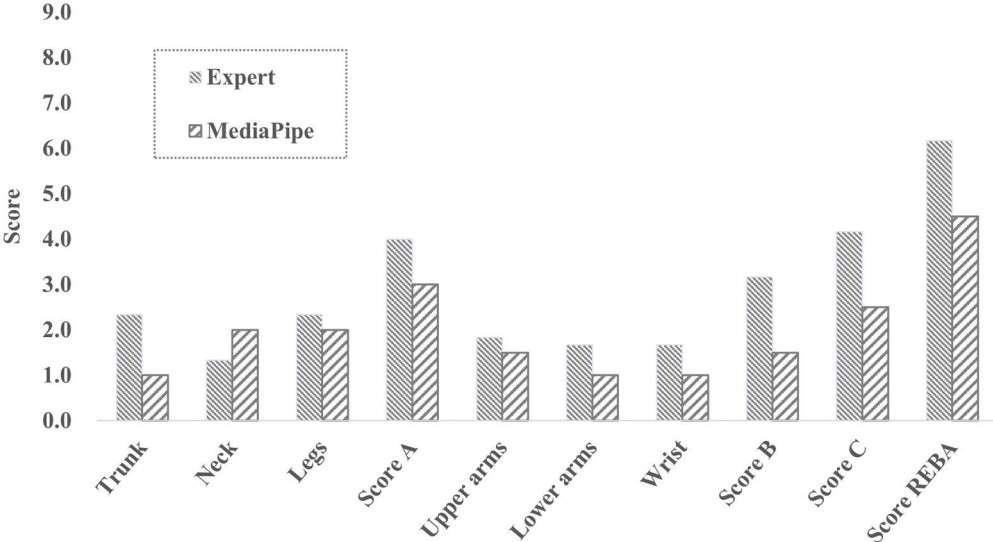

**Figure 35.** Comparison of cutter working posture scores.

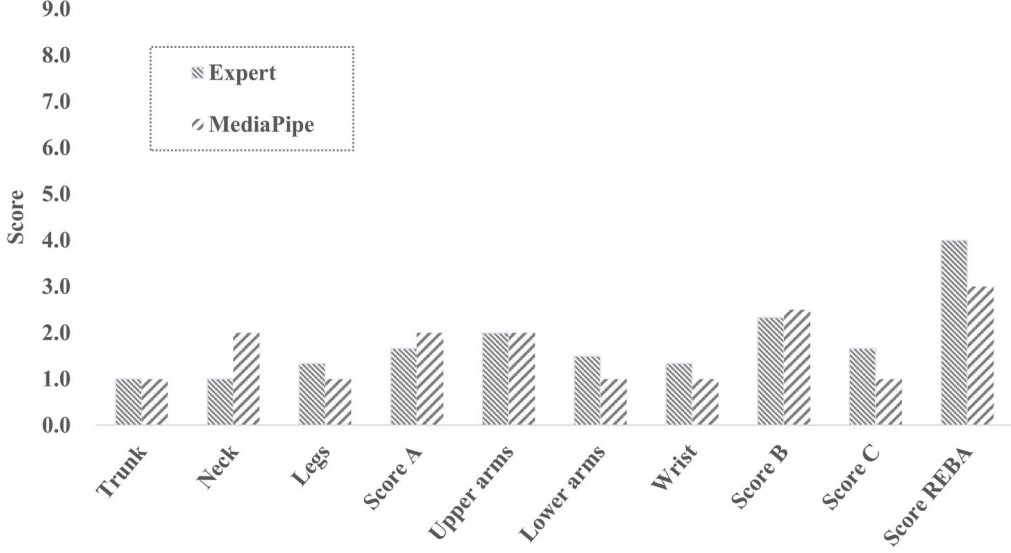

**Figure 36.** Comparison of circular saw working posture scores.

As a result of the comparison of the saw working posture scores (Figure 38), scores for each joint were similar overall, with a maximum error of one point, and MediaPipe scores were higher in all parts except for the trunk. The REBA scores were six points and five points, indicating the same action level (Level 2).

As a result of the comparison of the chisel working posture scores (Figure 39), scores for each joint were similar overall, with a maximum error of one point, and MediaPipe scores were higher in all parts, except for the upper and lower arms. The REBA scores were five points and six points, indicating the same action level (Level 2).

As a result of the comparison of the planar working posture scores (Figure 40), scores for each joint were similar overall, with a maximum error of 1.3 points, and the REBA scores were all 6 points, indicating the same action level (Level 2).

As a result of the comparison of the scores of the heavy load working posture (Figure 41), the scores for each joint were similar, with a maximum error of 1.3 points, and the REBA scores were all 10–11, indicating the same action level (Level 4).

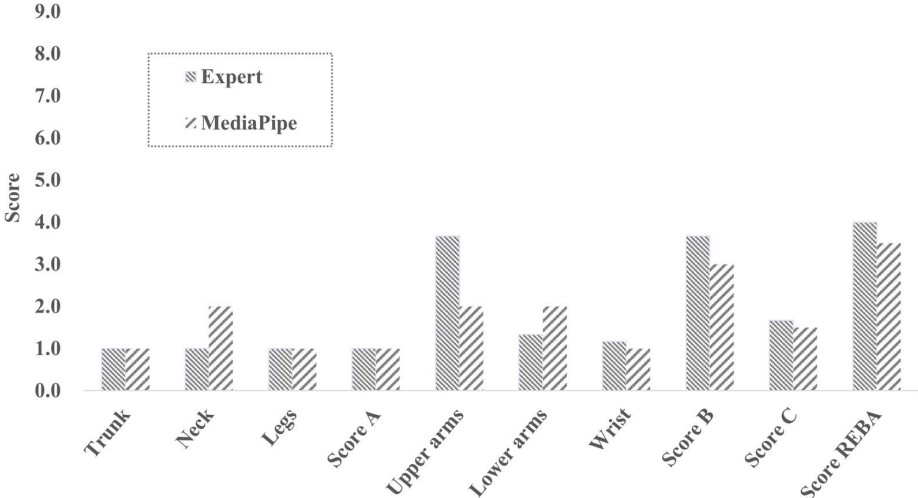

**Figure 37.** Comparison of drilling machine working posture scores.

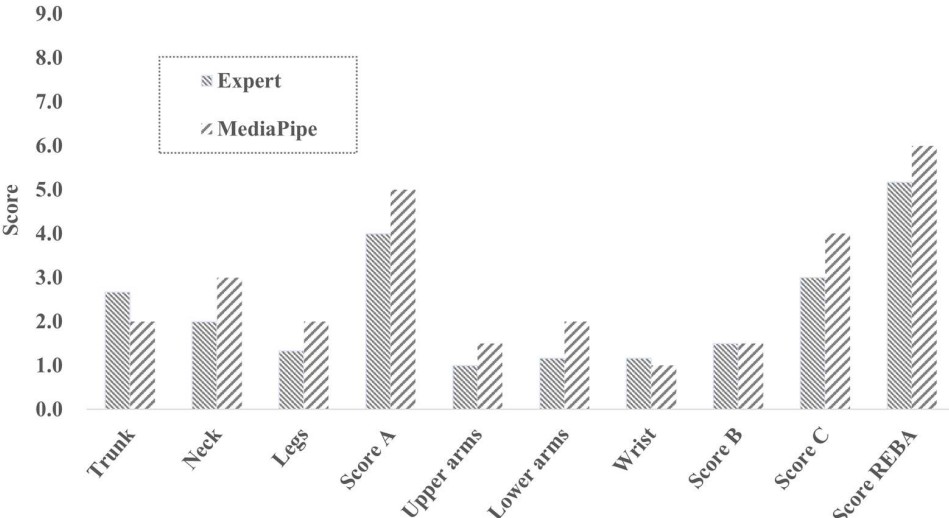

**Figure 38.** Comparison of saw working posture scores.

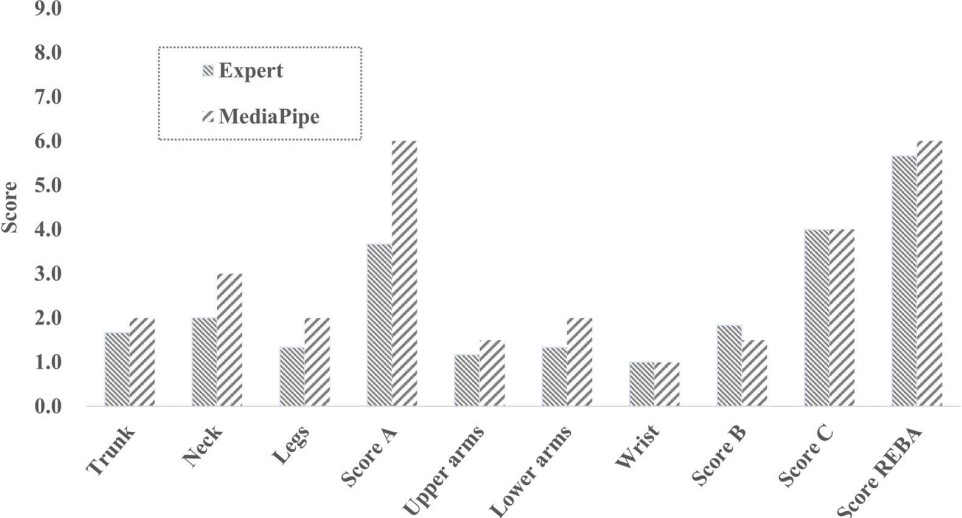

**Figure 39.** Comparison of chisel working posture scores.

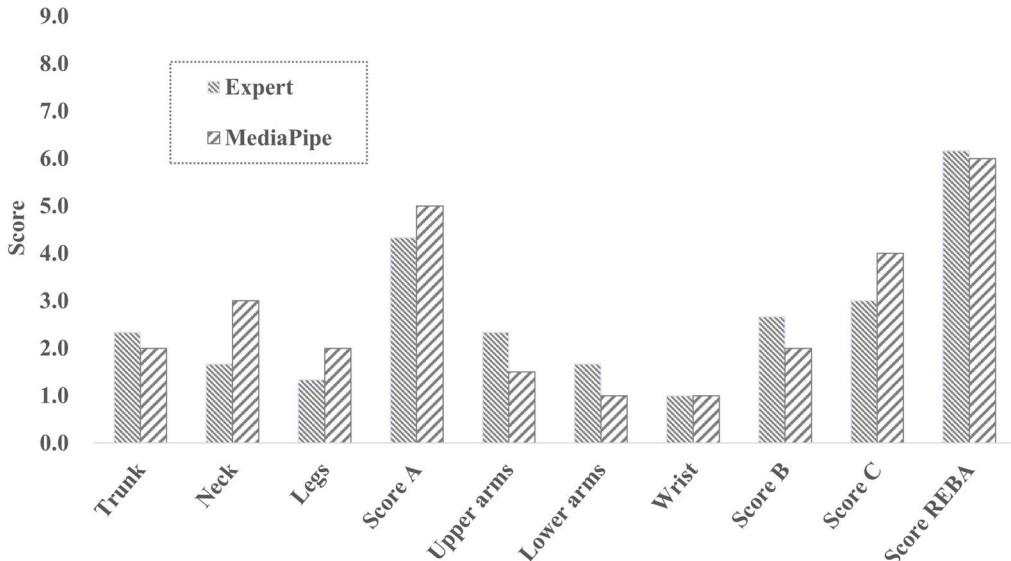

**Figure 40.** Comparison of planer working posture scores.

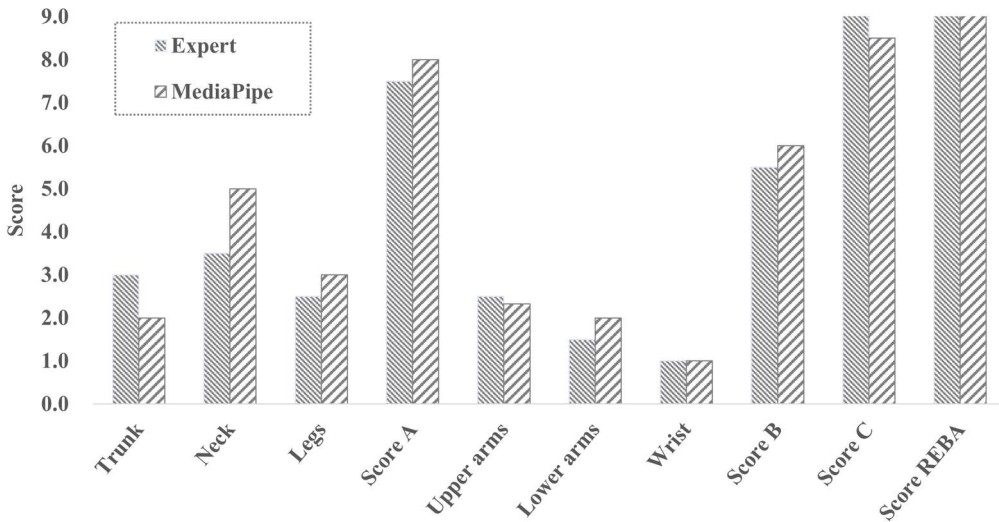

**Figure 41.** Comparison of heavy load working posture scores.

As a result of comparing the difference in measurement and evaluation time, it took an average of 40 min to evaluate all six working postures in the precision evaluation by ergonomic experts, whereas the precision evaluation using MediaPipe was performed in real time. Next, a comparison of the evaluation score of the ergonomic experts with the evaluation scores in this system shows that the overall scores were the same or with a slight error, but the musculoskeletal characteristics of each posture were the same. However, in the case of the cutting machine, the left arm was judged to be a safer angle than the right arm in the implemented system, but the left arm was judged to be a more dangerous angle than the right arm in the evaluation of ergonomic experts. Therefore, as a result of feedback, it was difficult to accurately identify the left arm because the left arm was covered during the experts' evaluation, and MediaPipe's results could be seen as more accurate.

## 5. Conclusions

The reports and research on domestic and foreign industrial sites show musculoskeletal disorder is a significant problem that affects not only economic and time loss, but also production efficiency. In this study, a method to measure the working posture of a

worker using MediaPipe's posture tracking technology and to evaluate the working posture ergonomically by applying the REBA technique was proposed.

First, the joint landmark was obtained by measuring the worker's working posture using MediaPipe, and the joint landmark data was applied to the 3D skeleton model and visualized in a virtual environment. Using the relative positions of each joint landmark, the angles of the joints required by the REBA evaluation technique were calculated and scores were given. In addition, a system implemented with the same working posture image was compared with the evaluation results of ergonomic experts, and it was found that there was a difference in the occlusion.

However, the joint angle visualization of MediaPipe is expected to be used to help the expert evaluation, thereby reducing the time and cost consumed for evaluation. In addition, it is expected that it can be useful in general industrial sites because it can determine the degree of load on the working posture.

However, due to the environmental characteristics of various industrial sites, recognition accuracy may be lowered, and it is difficult to apply in an image frame where a worker's face is covered. Therefore, future study aims to use images taken from multiple angles, using multiple cameras, and combined into one image to minimize the occlusion and increase the evaluation accuracy to.

It is expected that the ergonomic working method for the working posture analyzed through this study will reduce work effort and improve worker safety and efficiency.

**Author Contributions:** Conceptualization, S.-o.J. and J.K.; methodology, S.-o.J. and J.K.; software, S.-o.J.; validation, S.-o.J. and J.K.; formal analysis, S.-o.J.; investigation, S.-o.J.; resources, S.-o.J.; data curation, S.-o.J. and J.K.; writing—original draft preparation, S.-o.J. and J.K.; writing—review and editing, J.K.; visualization, S.-o.J.; supervision, J.K.; project administration, J.K.; funding acquisition, J.K. All authors have read and agreed to the published version of the manuscript.

**Funding:** This research received no external funding.

**Institutional Review Board Statement:** Not applicable.

**Informed Consent Statement:** Not applicable.

**Data Availability Statement:** Not applicable.

**Acknowledgments:** This research was funded by a 2021 research Grant from Sangmyung University.

**Conflicts of Interest:** The authors declare no conflict of interest.

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
