# Peer review of "CREBAS: Computer-Based REBA Evaluation System for Wood Manufacturers Using MediaPipe"

_applsci, doi:10.3390/app13020938_

Round 1

Reviewer 1 Report

Check that the style of writing is in the third person throughout. Don’t use ‘we’.

Check that the abstract provides an accurate synopsis of the paper. It is very vague in its present form.

Define abbreviations such as REBA or RULA when appears first in the paper. Don’t use abbreviations and their definitions everywhere in the paper. There are many such words throughout the paper.

Figures 1, 2, and 3 need citations as they are referred from some papers/surveys, etc. The same is true for the other few figures.

Too many results are presented unnecessarily. Please emphasize what is essential.

Hyperparameters of CNN can be summarised in tabular form.

Comment on computational time and complexity of the proposed method.

Was the data normalized/ standardized?

The authors do not mention the availability of this platform or framework for other practitioners, which is mandatory for this type of application. Add a comparison table at the end of the results section.

Author Response

Thank you for your thoughtful review.

The revisions are summarized in the attached document.

Reviewer 2 Report

In this article, the authors propose a computerised REBA assessment system (CREBAS) for musculoskeletal disorders in manufacturing industry using MediaPipe. They focused their study on the woodworking industry, a sector where workers have to perform repetitive physical tasks that require a lot of effort and can lead to injuries or other medical problems. The authors examined the technologies used to assess musculoskeletal risks, comparing and mentioning their characteristics and reliability.

The research methodology is clearly described; the experimental results are analysed by REBA experts and with the proposed system, called MediaPipe.

I kindly recommend to revise the following issues:

1. page 25..... the subtitle..."REBA evaluaiton with Mediapipe".. evaluation.....

2. there were analysed seven type of working postures (cutters, circular saws, drilling machines, saws, chisels, planers, and heavy loads) in both ways (experts method and MediaPipe) and at page 30 (section 4.4) is written "Table 17 shows the average scores of three evaluators for 6 types of work postures". Table 17 and 18 are presenting seven specific activities for wood industry. 

The proposed method brings important advantages in the evaluation of musculoskeletal disorders. I recommend mentioning some possible further research or applications of the proposed method (an example: ergonomic working methods for the analysed working postures that can reduce work effort and improve workers' safety and efficiency).

Author Response

(The authors gave the same response as above.)

Round 2

Reviewer 1 Report

The authors have addressed my comments.